# Uni-LoRA: One Vector is All You Need

**Kaiyang Li**
School of Computing
University of Connecticut
Storrs, CT 06269
kaiyang.li@uconn.edu

**Shaobo Han**
Optical Networking and Sensing
NEC Labs America
Princeton, NJ 08540
shaobo@nec-labs.com

**Qing Su**
School of Computing
University of Connecticut
Storrs, CT 06269
qing.2.su@uconn.edu

**Wei Li**
Dept. of Computer Science
Georgia State University
Atlanta, GA 30303
wli28@gsu.edu

**Zhipeng Cai**
Dept. of Computer Science
Georgia State University
Atlanta, GA 30303
zcai@gsu.edu

**Shihao Ji**
School of Computing
University of Connecticut
Storrs, CT 06269
shihao.ji@uconn.edu

## Abstract

Low-Rank Adaptation (LoRA) has become the de facto parameter-efficient fine-tuning (PEFT) method for large language models (LLMs) by constraining weight updates to low-rank matrices. Recent works such as Tied-LoRA, VeRA, and VB-LoRA push efficiency further by introducing additional constraints to reduce the trainable parameter space. In this paper, we show that the parameter space reduction strategies employed by these LoRA variants can be formulated within a unified framework, **Uni-LoRA**, where the LoRA parameter space, flattened as a high-dimensional vector space $\mathbb{R}^D$, can be reconstructed through a projection from a subspace $\mathbb{R}^d$, with $d \ll D$. We demonstrate that the fundamental difference among various LoRA methods lies in the choice of the projection matrix, $P \in \mathbb{R}^{D \times d}$. Most existing LoRA variants rely on layer-wise or structure-specific projections that limit cross-layer parameter sharing, thereby compromising parameter efficiency. In light of this, we introduce an efficient and theoretically grounded projection matrix that is isometric, enabling global parameter sharing and reducing computation overhead. Furthermore, under the unified view of Uni-LoRA, this design requires only a single trainable vector to reconstruct LoRA parameters for the entire LLM – making Uni-LoRA **both a unified framework and a "one-vector-only" solution**. Extensive experiments on GLUE, mathematical reasoning, and instruction tuning benchmarks demonstrate that Uni-LoRA achieves state-of-the-art parameter efficiency while outperforming or matching prior approaches in predictive performance. Our code is available at https://github.com/KaiyangLi1992/Uni-LoRA.

## 1 Introduction

Parameter-efficient fine-tuning (PEFT) [1] casts a new paradigm that leverages strong prior knowledge built in foundation models and adapts them to a wide range of downstream tasks by updating a small amount of trainable parameters. Among various PEFT methods, LoRA [2] has been particularly prevalent in recent studies. Given a pre-trained matrix $W_0 \in \mathbb{R}^{m \times n}$, LoRA constrains the weight increment $\Delta W$ as a low-rank decomposition $\Delta W = BA$, where $B \in \mathbb{R}^{m \times r}$ and $A \in \mathbb{R}^{r \times n}$ are trainable parameters, with $r \ll \min(m, n)$. LoRA reduces the fine-tuning cost significantly, while achieving impressive predictive performance.

To further reduce the number of trainable parameters for fine-tuning, recent methods augment [3, 4, 5] or modify [6] the LoRA architecture and tighten the trainable parameter space of LoRA into a lower-dimensional subspace, in which the fine-tuning is performed. For example, Tied-LoRA [3] ties all the $B$ and $A$ matrices of different LoRA-adapted modules and optimizes one pair of $B$ and

$A$ matrices and the diagonal entries of two diagonal matrices per LoRA module. VeRA [4] further reduces the trainable parameter space by randomly initializing $B$ and $A$ and freezing them afterwards, leading to fine-tuning only two vectors per LoRA module. LoRA-XS [5] introduces an $r \times r$ matrix per LoRA module, where $r$ is the pre-defined LoRA rank, and fine-tunes a set of $r \times r$ matrices. VB-LoRA [6] decomposes the $B$ and $A$ matrices of LoRA into fixed-length sub-vectors, and learns a globally shared vector bank and the compositional coefficients for each sub-vector of $B$s and $A$s. Despite the similarities among different LoRA variants, there isn't a unified framework of LoRA that can describe the aforementioned LoRA methods in a uniform language and enables a systematic analysis of all of them.

Inspired by the works of measuring the intrinsic dimension of objective landscapes [7, 8], we view the parameter space of LoRA as a $D$-dimensional space $\mathbb{R}^D$, and each LoRA variant defines a $d$-dimensional subspace $\mathbb{R}^d$, and a projection matrix $P \in \mathbb{R}^{D \times d}$ maps a trainable vector from the subspace $\mathbb{R}^d$ back to the full LoRA parameter space $\mathbb{R}^D$. From the perspective of this unified view, we recognize that most existing LoRA variants (e.g., Tied-LoRA, VeRA, and LoRA-XS) project the parameters of each LoRA module into a separate and fixed-dimensional subspace. However, recent studies (e.g., AdaLoRA [9] and LoRA-Drop [10]) have shown that the importance of LoRA modules varies across layers, suggesting that locally projecting the parameters of different LoRA modules into subspaces of the same dimensionality may be suboptimal. Moreover, our study reveals that the projection matrices $P$ used *implicitly* by Tied-LoRA, VeRA, and VB-LoRA do not possess the property of isometry [11] from the trainable parameter space to the original LoRA parameter space. In other words, those projections do not preserve the distance between parameter vectors in the original LoRA parameter space, distorting the geometry of the optimization landscape.

To address the aforementioned limitations, we propose **Uni-LoRA**, a unified framework of LoRA that treats the LoRA parameter space as a high-dimensional vector space and performs a *global* projection into a shared low-dimensional subspace. We further introduce an efficient and theoretically grounded projection matrix $P \in \mathbb{R}^{D \times d}$, in which each row is a one-hot vector with the index of "1" sampled uniformly from $d$ slots, followed by a column-wise normalization. As discussed in Sec. 3.3, this construction ensures that $P$ possesses three desirable properties for adaptation: globality, uniformity/load-balancing, and isometry. Conceptually, this corresponds to randomly partitioning all the $D$ parameters of LoRA into $d$ groups and enforce the parameters within each group to share the same value during the training process. We prove that the resulting projection matrix is isometric (or distance-preserving). Empirically, when fine-tuning the GEMMA-7B model, our Uni-LoRA achieves performance comparable to LoRA while training only 0.52M parameters – only **0.0061%** of the base model size and **0.26%** of the LoRA parameter size. Across multiple benchmarks, our method achieves extreme parameter efficiency, while outperforming or matching the state-of-the-art LoRA variants. Moreover, we show that our projection matrix attains the performance of the Fastfood projection [12]. While Fastfood is a widely used structured projection method with a time complexity of $\mathcal{O}(D \log d)$, our method achieves a significantly lower time complexity of $\mathcal{O}(D)$. Our contributions are summarized as follows:

- We propose a unified framework to analyze various LoRA variants, and show that many of them (e.g., Tied-LoRA, VeRA, LoRA-XS, and VB-LoRA) can be interpreted as projecting trainable parameters from the full LoRA parameter space into structured low-dimensional subspaces.

- We propose a simple yet extremely effective projection matrix that randomly partitions the LoRA parameters into equally sized groups and enforces all the parameters in each group to share the same value. Despite its simplicity, our approach outperforms or matches the performance of state-of-the-art LoRA methods across a wide range of tasks, including natural language understanding, mathematical reasoning, and instruction tuning.

- We further prove that our projection matrix is isometric and matches the performance of classical Fastfood projection in practice, while incurring significantly lower computational cost.

## 2  Related Work

**Parameter-efficient LoRA Variants.** Despite the success of Low-Rank Adaptation (LoRA) [2] in enabling parameter-efficient fine-tuning, several challenges remain – particularly when scaling to even larger models or deploying multiple adapters on resource-constrained mobile devices. Recent methods aim to further reduce the number of trainable parameters in LoRA by (1) selectively freezing a subset of the parameters [4, 5]; (2) enforcing weight-tying across layers [3]; (3) learning global parameter sharing through admixture reparameterization [6]. Our Uni-LoRA provides a unified

framework in which various parameter-reduction strategies are manifested as structural patterns in the projection matrix, thereby allowing the parameter redundancy in the LoRA parameter space to be studied and reduced through the lens of dimensionality reduction. Perhaps surprisingly, parameter sharing does not need to be layer-wise, structure-aware, or even learned. Our Uni-LoRA simplifies the fine-tuning with a random weight-sharing strategy, where only one parameter vector needs to be trained and stored, showcasing that one vector is all we need for LoRA.

**Intrinsic Dimension.** A growing line of work suggests that the effective degrees of freedom required to train machine learning models [7] and fine-tuning [8] lie in a significantly smaller subspace than the full parameter space. Zhang et al. [13] further demonstrate that the fine-tuning process tends to uncover task-specific intrinsic subspaces, and that disabling these subspaces severely harms generalization. Along this line, FourierFT [14] locally projects the layer-wise incremental parameter matrices (i.e., $\Delta W$) of pretrained models onto fixed Fourier bases, and shows that it suffices to only learn the corresponding combination coefficients.

Previous works [7, 8] typically employ distance-preserving dense Gaussian matrices or structured transforms such as Fastfood to project the original parameter space into a lower-dimensional subspace. In Uni-LoRA, we propose a novel construction of the projection matrix that is distance-preserving (i.e., isometric) while significantly reducing computational cost compared to dense Gaussian or Fastfood-based methods.

## 3 Proposed Method

### 3.1 Preliminaries: LoRA and its Parameter-efficient Variants

**LoRA.** LoRA [2] is a parameter-efficient fine-tuning technique for LLMs. Given a pre-trained weight matrix $W_0 \in \mathbb{R}^{m \times n}$, LoRA constrains the weight increment $\Delta W$ as a low-rank decomposition: $\Delta W = BA$, where $B \in \mathbb{R}^{m \times r}$ and $A \in \mathbb{R}^{r \times n}$ are trainable, low-rank matrices with $r \ll \min(m, n)$, which significantly reduce the number of trainable parameters and improve fine-tuning efficiency.

**Tied-LoRA / VeRA.** Tied-LoRA [3] and VeRA [4] augment the LoRA architecture with two extra diagonal matrices. Both methods constrain the weight increment as $\Delta W = \Lambda_b P_B \Lambda_d P_A$, where $P_B \in \mathbb{R}^{m \times r}$ and $P_A \in \mathbb{R}^{r \times n}$ are shared/tied across all LoRA modules, and $\Lambda_b \in \mathbb{R}^{m \times m}$ and $\Lambda_d \in \mathbb{R}^{r \times r}$ (per LoRA module) are trainable diagonal matrices that can selectively enable or disable columns and rows of $P_B$ and $P_A$ by scaling, allowing effective, fine-grained adaptation with a minimal number of trainable parameters. The main difference between Tied-LoRA and VeRA is that while $P_B$ and $P_A$ are trainable in Tied-LoRA, they are randomly initialized and frozen in VeRA, leading to a higher parameter efficiency.

**VB-LoRA.** VB-LoRA [6] decomposes the $B$ and $A$ matrices of LoRA into fixed-length sub-vectors, and learns a globally shared vector bank and the compositional coefficients to generate each sub-vector of $B$s and $A$s. During fine-tuning, only the parameters in the vector bank and the compositional coefficients for each sub-vector are trained. Upon the completion of fine-tuning, only the vector bank and the indices and values of the top-$K$ (e.g., $K = 2$) compositional coefficients are stored, leading to extremely small stored parameter size.

### 3.2 Uni-LoRA: A Unified Framework of LoRA

Despite the differences among the aforementioned LoRA variants, all of them augment or modify the LoRA architecture and tighten the trainable parameter space of LoRA into a lower-dimensional subspace. In this paper, we show that all these parameter space reduction strategies can be formulated within a unified framework, which enables a systematic analysis of all these LoRA variants.

Inspired by the works on measuring the intrinsic dimension of objective landscapes [7, 8], we view the parameter space of LoRA as a $D$-dimensional space $\mathbb{R}^D$, and each LoRA variant defines a $d$-dimensional subspace $\mathbb{R}^d$, and a projection matrix $P \in \mathbb{R}^{D \times d}$ maps a trainable vector from the subspace $\mathbb{R}^d$ back to the full LoRA parameter space $\mathbb{R}^D$. Specifically, for each LoRA-adapted module $\ell = 1, \cdots, L$, we row-flatten the low-rank matrices $B^\ell \in \mathbb{R}^{m \times r}$ and $A^\ell \in \mathbb{R}^{r \times n}$, and concatenate them to construct a full parameter vector of LoRA:

$$\theta_D = \text{Concat}\big(\text{vec}_{\text{row}}(B^1), \text{vec}_{\text{row}}(A^1), \cdots, \text{vec}_{\text{row}}(B^L), \text{vec}_{\text{row}}(A^L)\big), \tag{1}$$

where $\text{vec}_{\text{row}}(\cdot)$ denotes the row-wise flatten of a matrix into a column vector, and $D = L(m + n)r$ is the total number of LoRA parameters. Instead of optimizing directly in the LoRA parameter space,

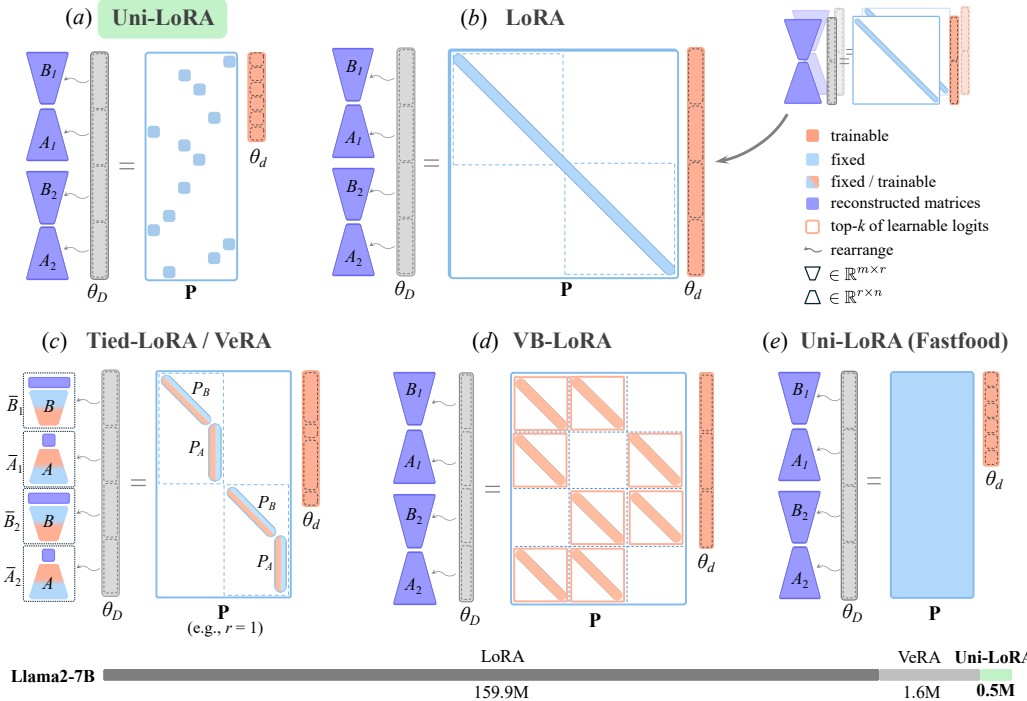

Figure 1: Overview of Uni-LoRA and the representations of various LoRA methods in the unified framework. For better visualization, we illustrate the framework with only two LoRA-adapted modules. More examples including LoRA-XS and FourierFT are provided in Appendix A.1.

we propose to work in a lower-dimensional subspace with a parameter vector $\theta_d \in \mathbb{R}^d$, which is projected to the original LoRA parameter space by a linear projection:

$$\theta_D = P\theta_d, \tag{2}$$

where $P \in \mathbb{R}^{D \times d}$ is a projection matrix with $d \ll D$. In this formulation, $\theta_d \in \mathbb{R}^d$ represents the trainable parameters, and $P$ can be trained along with $\theta_d$ or designed and frozen during the fine-tuning process. We show that this view unifies a broad class of parameter-efficient LoRA variants, including Tied-LoRA, VeRA, LoRA-XS, and VB-LoRA. Therefore, we call the formulation expressed by Eq. 2 Uni-LoRA, a unified framework of LoRA. In light of this unified framework, we demonstrate that the fundamental difference among various LoRA methods lies in the choice of the projection matrix $P$ and whether $P$ is trained along with $\theta_d$ or not.

Figure 1 illustrates the framework of Uni-LoRA and the representations of various LoRA methods, including LoRA, Tied-LoRA, VeRA, VB-LoRA, and Uni-LoRA (Fastfood) in this unified framework.

**Uni-LoRA** As a specific instantiation of this unified framework, Uni-LoRA further introduces an efficient and theoretically grounded projection matrix $P \in \mathbb{R}^{D \times d}$, in which each row is a one-hot vector with the index of "1" sampled uniformly from $d$ slots, followed by a column-wise normalization such that the nonzero entries in column $j$ are set to $1/\sqrt{n_j}$, where $n_j$ denotes the number of nonzero entries in that column. Once initialized in such a way, $P$ **remains frozen and only the parameter vector $\theta_d$ is fine-tuned** and projected back to the full LoRA parameter space $\theta_D$ by $P$. Conceptually, this corresponds to randomly partitioning all the $D$ parameters of LoRA into $d$ groups, with parameters within each group constrained to share the same value during fine-tuning. Theorem 1 shows that such designed projection matrix $P$ is isometric, which preserves the distance between parameter vectors in the original LoRA parameter space, without distorting the geometry of the optimization landscape.

Algorithm 1 provides the PyTorch-like pseudocode for Uni-LoRA, which can be seamlessly integrated into the PyTorch framework. Given that the projection matrix $P$ is one-hot like matrix, $P$ isn't explicitly constructed. Instead, only the indices and values of nonzero entries of this sparse matrix involve in the computation, leading to extremely efficient implementation.

**LoRA** It is straightforward to represent LoRA in the framework of Uni-LoRA. As illustrated in Figure 1(b), in this case, $P$ corresponds to a $D \times D$ identity matrix, and $\theta_d$ has the same dimensionality

of the original LoRA parameter space, i.e., $d = D$. Apparently, the identity matrix $P$ is isometric, but it doesn't have the effect of reducing number of trainable parameters of LoRA.

**Tied-LoRA / VeRA**    As introduced in Section 3.1, Tied-LoRA [3] and VeRA [4] represents the weight increment as $\Delta W = \Lambda_b P_B \Lambda_d P_A$, where $P_B$ and $P_A$ are shared/tied across all the LoRA-adapted modules, and $\Lambda_b$ and $\Lambda_d$ are defined per LoRA module. Specifically, for each LoRA module $\ell = 1, \cdots, L$, we extract the diagonal elements of $\Lambda_b^\ell \in \mathbb{R}^{m \times m}$ and $\Lambda_d^\ell \in \mathbb{R}^{r \times r}$, and concatenate them to construct a trainable parameter vector:

$$\theta_d = \text{Concat}\left(\text{diag}(\Lambda_b^1), \text{diag}(\Lambda_d^1), \cdots, \text{diag}(\Lambda_b^L), \text{diag}(\Lambda_d^L)\right), \tag{3}$$

where $\text{diag}(\cdot)$ denotes the diagonal vector of a matrix, and $d = L(m + r)$ is the number of trainable parameters of Tied-LoRA or VeRA. On the other hand, the full parameter vector of LoRA can be formulated as

$$\theta_D = \text{Concat}\left(\text{vec}_{\text{row}}(\bar{B}^1), \text{vec}_{\text{row}}(\bar{A}^1), \cdots, \text{vec}_{\text{row}}(\bar{B}^L), \text{vec}_{\text{row}}(\bar{A}^L)\right), \tag{4}$$

where $\bar{B}^\ell = \Lambda_b^\ell P_B$ and $\bar{A}^\ell = \Lambda_d^\ell P_A$, and they correspond to the LoRA parameters $B^\ell$ and $A^\ell$.

As illustrated in Figure 1(c), the projection matrix $P$ of Tied-LoRA and VeRA exhibits a structured sparse pattern, composed of block-diagonal components reshaped from $P_B$ and $P_A$ (details of which are relegated to Appendix A.1). Since $P_B$ and $P_A$ are shared/tied cross all the LoRA modules, the block-diagonal components are repeated $L$ times in $P$. The main difference between Tied-LoRA and VeRA is the trainability of $P_B$ and $P_A$, i.e., the projection matrix $P$ is trainable in Tied-LoRA and frozen in VeRA. This is indicated by two different colors in the diagram.

It is worth noting that the projection matrix $P$ in Tied-LoRA and VeRA is structured to have non-zero entries only in diagonal blocks, suggesting that the projection is local in nature. Also, the rows of the LoRA matrices $\bar{B}^\ell \in \mathbb{R}^{m \times r}$ and $\bar{A}^\ell \in \mathbb{R}^{r \times n}$ are projected to two separate lower-dimensional subspaces with the non-uniform dimensionalities ($m$ vs. $r$). Moreover, this projection matrix is not isometry, indicating that it may distort the geometric structure of the original parameter space. Collectively, the locality, non-uniformity, and non-isometric nature of this projection may constrain adaptation flexibility and limit representational expressiveness.

**Uni-LoRA (Fastfood)**    This is a variant of Uni-LoRA, in which the project matrix $P$ is initialized with the classical structured Fastfood projection [12], which is isometric. Fastfood approximates dense Gaussian projections with structured transforms and reduces the time complexity of a naïve implementation from $\mathcal{O}(Dd)$ to $\mathcal{O}(D \log d)$. As to be discussed in Section 3.4, Uni-LoRA with our uniform random projection achieves a significantly lower time complexity of $\mathcal{O}(D)$.

Similarly, VB-LoRA, LoRA-XS, and FourierFT can be represented as specific instantiations in our unified framework. Due to the page limits, the details are relegated to Appendix A.1.

### 3.3    Analysis of the Projection Matrix

Our discussion in Section 3.2 reveals that the key distinctions among various LoRA methods lie in the choice of the projection matrix $P$ and whether $P$ is trained along with $\theta_d$ or not. We argue that a well-structured $P$ should have the following three properties, i.e., globality, uniformity/load-balanced, and isometry, which can substantially enhance adaptation performance.

- **Globality**: Global parameter sharing across different types of matrices and layers breaks the physical barrier and enables maximal reduction of parameter redundancy.

- **Uniformity / Load-Balanced**: Each subspace dimension is mapped to roughly equal number of dimensions of the original full parameter space, such that the information is evenly distributed to all subspace dimensions and load-balanced.

- **Isometry**: The projection preserves the distance between parameter vectors in the original full parameter space, without distorting the geometric structure of the optimization landscape.

According to the properties above, we characterize the projection matrices employed by various LoRA variants in Table 1. Theorem 1 proves that the uniform random projection introduced by our Uni-LoRA is isometric.

**Theorem 1.** *Let $P \in \mathbb{R}^{D \times d}$ be a projection matrix where each row selects exactly one entry uniformly at random to be nonzero, and sets all other entries to zero. Let $n_j$ denote the number of nonzero*

| Method | Learnable Projection | Globality | Uniformity | Isometry |
|---|---|---|---|---|
| VeRA [4] | ✗ | ✗ | ✗ | ✗ |
| TiedLoRA [3] | ✓ | ✗ | ✗ | ✗ |
| VB-LoRA [6] | ✓ | ✓ | ✓ | ✗ |
| LoRA-XS [5] | ✗ | ✗ | ✓ | ✓ |
| Uni-LoRA (Fastfood) | ✗ | ✓ | ✓ | ✓ |
| Uni-LoRA (ours) | ✗ | ✓ | ✓ | ✓ |

Table 1: Properties of the projection matrices $P$ employed by various LoRA methods, where "Learnable Projection" refers to besides $\theta_d$ whether $P$ itself contains trainable parameters.

entries in column $j$, and $n_j > 0$ [1]. *For column-wise normalization, each nonzero entry in column $j$ is set to* $1/\sqrt{n_j}$. *As a projection matrix, $P$ is isometric. Formally,* $\|P(x-y)\| = \|x-y\|, \forall x, y \in \mathbb{R}^d$.

*Proof.* Given the construction of $P$ in the theorem statement, we begin by showing that $P^\top P = I_d$. Consider the $(j,k)$-th entry of $P^\top P$: $[P^\top P]_{j,k} = \sum_{i=1}^{D} P_{i,j} P_{i,k}$. **Case 1:** $j \neq k$**.** Since each row contains only one nonzero entry, there exists no row $i$ such that both $P_{i,j}$ and $P_{i,k}$ are nonzero. Hence, every term in the summation is zero, and we have $[P^\top P]_{j,k} = 0$. **Case 2:** $j = k$**.** Column $j$ contains $n_j$ nonzero entries, each of value $1/\sqrt{n_j}$. Thus, $[P^\top P]_{j,j} = \sum_{i=1}^{D} P_{i,j}^2 = n_j \cdot \left(1/\sqrt{n_j}\right)^2 = 1$. From the analysis above, we have $[P^\top P]_{j,k} = 1$, if $j = k$ and 0 otherwise, thus $P^\top P = I_d$. Therefore,

$$\|P(x-y)\|^2 = (x-y)^\top P^\top P(x-y) = (x-y)^\top (x-y) = \|x-y\|^2, \forall x, y \in \mathbb{R}^d$$

This completes the proof. □

**Why existing LoRA variants may be suboptimal?** In light of our unified framework, Figure 1 reveals that the projections used by Tied-LoRA, VeRA, and LoRA-XS (except VB-LoRA) are all local with layer-wise projection; furthermore, Tied-LoRA and VeRA are inherently non-uniform. Specifically, although the $B$ and $A$ matrices of LoRA contain the same number of parameters, Tied-LoRA and VeRA ties the parameters of $B$ with more parameters in the lower-dimensional subspace than $A$ (i.e., $m$ vs. $r$). Intuitively, such a non-uniform projection may be suboptimal as information of the full LoRA parameter space is non-uniformly distributed to lower-dimensional subspace. In Section 4.5, we conduct a controlled experiment comparing uniform and non-uniform projections, and the results confirm that the uniform project consistently outperforms its non-uniform counterpart.

**Why Isometry is important?** Prior works [15, 16] show that if the projection matrix is *isometric* (which satisfies $\|P(x-y)\| = \|x-y\|$ for any pair of vectors $x, y$), the geometry of the original space is preserved in the projected subspace. This property ensures that the optimization landscape remains faithful to the original parameter space, making it particularly well-suited for subspace training for neural networks.

### 3.4 Complexity Analysis

**Storage Complexity.** Since the projection matrix $P$ of Uni-LoRA is generated from a random seed, upon the completion of training, only the random seed and the learned subspace vector $\theta_d \in \mathbb{R}^d$ need to be stored or transmitted. As a result, the total number of stored data is only $d + 1$, leading to an extremely compact model representation and showcasing that one vector is all we need for LoRA.

**Time and Space Complexity of Projection.** The projection matrix $P$ of Uni-LoRA is a sparse matrix with exactly $D$ nonzero entries. As a result, the projection operation $P\theta_d$ has both time and space complexity of $\mathcal{O}(D)$. In contrast, the classical distance-preserving methods such as dense Gaussian projections require $\mathcal{O}(Dd)$ time and space, while the Fastfood transform [12] has a time complexity of $\mathcal{O}(D \log d)$ and a space complexity of $\mathcal{O}(D)$.

To summarize, our Uni-LoRA not only provides a unified framework of LoRA but also introduces an efficient and theoretically grounded projection, which enjoys all three desired properties that we discussed above. This approach requires no architectural modifications, no sparsity priors, and achieves a significantly lower time complexity than the classical Fastfood transform.

---

[1]To ensure the condition $n_j > 0$ is always held, we can re-sample $P$ if one column happens to be all zeros.

**Algorithm 1** Pseudocode of Uni-LoRA in a PyTorch-like Style

```
# in_dim, out_dim, r: input/output dimensions of the linear layer, LoRA rank
# theta_d: only trainable parameter vector of length d
# index_A/B: maps LoRA matrices A/B to subspace vector  theta_d
# norm_factor_A/B: normalization matrix, same shape as index_A/B, with entry value
# norm_factor_A[i,j]=1/sqrt(n_k), where k=index_A[i,j] and n_k is its occurrence
# frequency across all index_As and index_Bs.
# x, W: input and original weight
class Uni-LoRA:
def __init__(self, in_dim, out_dim, r, theta_d):
   d = len(theta_d)
   self.index_A = torch.randint(0, d, (in_dim, r), dtype=torch.long)
   self.index_B = torch.randint(0, d, (r, out_dim), dtype=torch.long)
   self.theta_d = theta_d

def update_norm_factor(self,norm_factor_A,norm_factor_B):
    self.norm_factor_A = norm_factor_A
    self.norm_factor_B = norm_factor_B

def forward(self,W,x):
    A = self.theta_d[self.index_A] * self.norm_factor_A
    B = self.theta_d[self.index_B] * self.norm_factor_B
    # For memory efficiency, we avoid explicitly computing \delta W = B @ A.
    return x @ W + (x @ A) @ B
```

# 4    Experiments

In this section, we evaluate our method through a series of experiments. We begin by comparing Uni-LoRA to the state-of-the-art PEFT methods: LoRA, Tied-LoRA, VeRA, LoRA-XS, and FourierFT on the GLUE benchmark. Next, we extend our analysis to mathematical reasoning tasks on Mistral and Gemma models, as well as instruction tuning tasks on Llama2. All our experiments are conducted on a server equipped with 8 NVIDIA A100 80GB GPUs. For reproducibility, we provide detailed hyperparameters and specifications of computing resources for each experiment in Appendix A.2.

## 4.1    Natural Language Understanding

We adopt the General Language Understanding Evaluation (GLUE) benchmark [17] to assess the performance of Uni-LoRA across various natural language understanding tasks. Following [4, 6], we focus on six tasks from GLUE: SST-2 [18] (sentiment analysis), MRPC [19] (paraphrase detection), CoLA [20] (linguistic acceptability), QNLI [21] (inference), RTE [22] (inference) and STS-B [23] (semantic textual similarity). Our experiments use RoBERTa$_{base}$ and RoBERTa$_{large}$ [24] as the backbone models. We apply LoRA with rank 4 to the query and value matrices in each Transformer layer, and project the resulting LoRA parameter space into a subspace of dimension $d = 23,040$, matching the subspace size used by the best-performing VB-LoRA baseline [6].

Table 2 reports the results across 12 cases spanning 6 GLUE tasks and 2 model scales. Uni-LoRA ranks either first or second in 11 out of the 12 cases, with the only exception being on the RTE task of RoBERTa$_{large}$, where the dataset is small and the performance variance is relatively high. As we can see, Uni-LoRA consistently outperforms Tied-LoRA, VeRA, LoRA-XS, and FourierFT while requiring fewer trainable parameters. For example, when fine-tuning RoBERTa$_{large}$, our method reduces the number of trainable parameters to less than 40% of that required by Tied-LoRA or VeRA, while achieving better performance across all tasks. Moreover, Uni-LoRA achieves performance comparable to VB-LoRA while using significantly fewer trainable parameters, suggesting that our fixed projection matrix is as effective as the learned projection in VB-LoRA. Furthermore, since our projection matrix is not learned, Uni-LoRA incurs substantially lower computational overhead than VB-LoRA (See Tables 9, 10, 12 in the appendix for details).

---

[1]† The original VB-LoRA paper reports the number of stored parameters rather than trainable ones: 0.23M for RoBERTa$_{base}$, 0.24M for RoBERTa$_{large}$, 0.65M for Mistral-7B, 0.67M for Gemma-7B, 0.8M for Llama2-7B, and 1.1M for Llama2-13B. For consistency, we report the number of trainable parameters throughout all tables, following the convention used in other baselines.

Table 2: Results with RoBERTa$_{base}$ and RoBERTa$_{large}$ on the GLUE benchmark. The best results in each group are shown in **bold**, while the second-best results are denoted with underlining. We report Matthew's correlation for CoLA, Pearson correlation for STS-B, and Accuracy for all other tasks. We report the median performance over 5 runs with different random seeds.

| | Method | # Trainable Params | SST-2 | MRPC | CoLA | QNLI | RTE | STS-B | Avg. |
|---|---|---|---|---|---|---|---|---|---|
| BASE | FT | 125M | 94.8 | 90.2 | 63.6 | 92.8 | 78.7 | 91.2 | 85.2 |
| | LoRA | 0.295M | $95.1_{\pm 0.2}$ | $89.7_{\pm 0.7}$ | $63.4_{\pm 1.2}$ | $93.3_{\pm 0.3}$ | $86.6_{\pm 0.7}$ | $91.5_{\pm 0.2}$ | 86.6 |
| | VeRA | 0.043M | $\mathbf{94.6}_{\pm 0.1}$ | $89.5_{\pm 0.5}$ | $\mathbf{65.6}_{\pm 0.8}$ | $91.8_{\pm 0.2}$ | $78.7_{\pm 0.7}$ | $\mathbf{90.7}_{\pm 0.2}$ | 85.2 |
| | Tied-LoRA | 0.043M | $94.4_{\pm 0.5}$ | $88.5_{\pm 1.0}$ | $61.9_{\pm 1.6}$ | $92.0_{\pm 0.4}$ | $76.2_{\pm 1.3}$ | $89.8_{\pm 0.3}$ | 83.8 |
| | VB-LoRA | 0.075M† | $94.4_{\pm 0.5}$ | $89.5_{\pm 0.5}$ | $63.3_{\pm 0.7}$ | $\underline{92.2}_{\pm 0.2}$ | $\mathbf{82.3}_{\pm 1.3}$ | $\underline{90.8}_{\pm 0.1}$ | **85.4** |
| | FourierFT | 0.024M | $94.2_{\pm 0.3}$ | $\mathbf{90.0}_{\pm 0.8}$ | $63.8_{\pm 1.6}$ | $\underline{92.2}_{\pm 0.1}$ | $79.1_{\pm 0.5}$ | $\underline{90.8}_{\pm 0.2}$ | 85.0 |
| | Uni-LoRA*(Ours)* | **0.023M** | $\underline{94.5}_{\pm 0.2}$ | $\underline{89.7}_{\pm 0.4}$ | $\underline{64.6}_{\pm 1.0}$ | $\mathbf{92.3}_{\pm 0.4}$ | $\underline{79.8}_{\pm 1.1}$ | $\mathbf{90.9}_{\pm 0.2}$ | $\underline{85.3}$ |
| LARGE | LoRA | 0.786M | $96.2_{\pm 0.5}$ | $90.2_{\pm 1.0}$ | $68.2_{\pm 1.9}$ | $94.8_{\pm 0.3}$ | $85.2_{\pm 1.1}$ | $92.3_{\pm 0.5}$ | 87.8 |
| | VeRA | 0.061M | $\underline{96.1}_{\pm 0.1}$ | $90.9_{\pm 0.7}$ | $68.0_{\pm 0.8}$ | $94.4_{\pm 0.2}$ | $85.9_{\pm 0.7}$ | $91.7_{\pm 0.8}$ | 87.8 |
| | Tied-LoRA | 0.066M | $94.8_{\pm 0.6}$ | $89.7_{\pm 1.0}$ | $64.7_{\pm 1.2}$ | $94.1_{\pm 0.1}$ | $81.2_{\pm 0.1}$ | $90.8_{\pm 0.3}$ | 85.9 |
| | VB-LoRA | 0.162M† | $\underline{96.1}_{\pm 0.2}$ | $\mathbf{91.4}_{\pm 0.6}$ | $\underline{68.3}_{\pm 0.7}$ | $\mathbf{94.7}_{\pm 0.5}$ | $86.6_{\pm 1.3}$ | $91.8_{\pm 0.1}$ | $\underline{88.2}$ |
| | LoRA-XS | 0.025M | $95.9_{\pm 0.3}$ | $90.7_{\pm 0.4}$ | $67.0_{\pm 1.2}$ | $93.9_{\pm 0.1}$ | $\mathbf{88.1}_{\pm 0.3}$ | $\underline{92.0}_{\pm 0.1}$ | 87.9 |
| | FourierFT | 0.048M | $96.0_{\pm 0.2}$ | $90.9_{\pm 0.3}$ | $67.1_{\pm 1.4}$ | $94.4_{\pm 0.4}$ | $\underline{87.4}_{\pm 1.6}$ | $\underline{92.0}_{\pm 0.4}$ | 88.0 |
| | Uni-LoRA*(Ours)* | **0.023M** | $\mathbf{96.3}_{\pm 0.2}$ | $\underline{91.3}_{\pm 0.6}$ | $\mathbf{68.5}_{\pm 1.1}$ | $\underline{94.6}_{\pm 0.4}$ | $86.6_{\pm 1.6}$ | $\mathbf{92.1}_{\pm 0.1}$ | **88.3** |

Table 3: Results with the Mistral-7B and Gemma-7B models on the GSM8K and MATH Benchmarks. The best results in each group are shown in **bold**. Baselines' results are either reproduced using their reported configurations or directly sourced from their original papers.

| Model | Method | # Parameters | GSM8K | MATH |
|---|---|---|---|---|
| MISTRAL-7B | Full-FT | 7242M | 67.02 | 18.60 |
| | LoRA | 168M | 67.70 | 19.68 |
| | LoRA-XS | 0.92M | 68.01 | 17.86 |
| | VB-LoRA | 93M† | **69.22** | 17.90 |
| | VeRA | 1.39M | 68.69 | **18.81** |
| | FourierFT | 0.67M | 68.92 | 17.50 |
| | Uni-LoRA *(Ours)* | **0.52M** | 68.54 | 18.18 |
| GEMMA-7B | Full-FT | 8538M | 71.34 | 22.74 |
| | LoRA | 200M | 74.90 | 31.28 |
| | LoRA-XS | 0.80M | 74.22 | 27.62 |
| | VB-LoRA | 113M† | 74.86 | 28.90 |
| | VeRA | 1.90M | 74.98 | 28.84 |
| | FourierFT | 0.59M | 72.97 | 25.14 |
| | Uni-LoRA *(Ours)* | **0.52M** | **75.59** | **28.94** |

Table 4: Score$_1$ and Score$_2$ denote evaluations on MT-Bench single-turn and multi-turn dialogues. LoRA* and VeRA were scored using an older GPT-4 API. LoRA# and other methods were evaluated using the updated version. The two sets of results are reported separately for fairness.

| Model | Method | # Parameters | Score$_1$ | Score$_2$ |
|---|---|---|---|---|
| LLAMA2 7B | LoRA* | 159.9M | 5.19 | - |
| | VeRA | 1.6M | 5.08 | - |
| | w/o FT | - | 1.31 | 1.11 |
| | LoRA# | 159.9M | **5.62** | 3.23 |
| | VBLoRA | 83M† | 5.43 | $\underline{3.46}$ |
| | Uni-LoRA *(Ours)* | **0.52M** | $\underline{5.58}$ | **3.56** |
| LLAMA2 13B | LoRA* | 250.3M | 5.77 | - |
| | VeRA | 2.4M | 5.93 | - |
| | w/o FT | - | 1.46 | 1.06 |
| | LoRA# | 250.3M | $\underline{6.20}$ | 4.13 |
| | VBLoRA | 256M† | 5.96 | $\underline{4.33}$ |
| | Uni-LoRA *(Ours)* | **1.0M** | **6.34** | **4.43** |

## 4.2 Mathematical Reasoning

To evaluate mathematical reasoning capabilities, we fine-tune the Mistral-7B-v0.1 [25] and Gemma-7B [26] models on the MetaMathQA [27] dataset and test them on GSM8K [28] and MATH [29]. We compare our results with the state-of-the-art methods, following their experimental configurations. Table 3 shows that Uni-LoRA achieves strong and consistent performance across both Mistral-7B and Gemma-7B backbones. On the MATH benchmark, it reaches 18.18 with Mistral-7B and 28.94 with Gemma-7B, outperforming LoRA-XS, VB-LoRA and FourierFT. It also yields higher accuracy than LoRA-XS on both GSM8K and MATH, while using less than 70% of its trainable parameters. Compared to VB-LoRA, Uni-LoRA achieves similar or better performance with substantially fewer parameters. It also demonstrates comparable accuracy to VeRA on mathematical reasoning tasks, while requiring only half as many trainable parameters on Mistral-7B and less than one-third on Gemma-7B. Furthermore, Uni-LoRA consistently outperforms FourierFT on both tasks, while being more parameter-efficient.

## 4.3 Instruction Tuning

Instruction tuning refers to the process of fine-tuning a model on a set of instructions or prompts to improve its ability to follow task-specific directives [30]. We perform instruction tuning on the Llama 2 [31] model using the Cleaned Alpaca dataset [32], which improves the quality of the original Alpaca dataset. Following [4, 6], we fine-tune the Llama 2 model [31] within the QLoRA framework [33], which enables low-memory fine-tuning of LLMs on a single GPU. The fine-tuned models generate

responses to MT-Bench questions, which are then scored by GPT-4 on a 10-point scale. We apply Uni-LoRA to project the rank-4 LoRA weights into a 0.5M-dimensional subspace for Llama2-7B and a 1M-dimensional subspace for Llama2-13B.

Table 4 reports the results. Due to a noticeable discrepancy between the reported scores, we include two sets of LoRA results for each experiment: one from Kopiczko et al. [4] and another from our own reimplementation. Although we closely follow the experimental settings of Kopiczko et al. [4], we speculate that the difference may be attributed to changes in the GPT-4 model over time. Nevertheless, comparing the relative performance of VeRA and Uni-LoRA with respect to their corresponding LoRA baselines remains meaningful. As we can see, Uni-LoRA achieves performance comparable to LoRA while using only 0.3% (Llama2-7B) and 0.4% (Llama2-13B) of the LoRA parameters. Similarly, VeRA also demonstrates comparable performance relative to its own LoRA baseline, but it requires more than twice the number of trainable parameters compared to Uni-LoRA. Furthermore, Uni-LoRA consistently outperforms VB-LoRA across both evaluation metrics on Llama2-7B and Llama2-13B, while using significantly fewer trainable parameters of VB-LoRA and slightly fewer stored parameters.

## 4.4 Experiments of Computer Vision Tasks

To further assess the effectiveness of Uni-LoRA on more complex vision tasks, we follow the experimental setup in FourierFT [14] and conduct additional evaluations. Specifically, we apply Uni-LoRA with $d = 74,000$ on ViT-Base and $d = 144,000$ on ViT-Large, using a fixed rank $r = 4$. A grid search is conducted over the head learning rate in $\{1 \times 10^{-3}, 2 \times 10^{-3}, 5 \times 10^{-3}, 1 \times 10^{-2}\}$ and the trainable vector learning rate in $\{2 \times 10^{-3}, 5 \times 10^{-3}, 1 \times 10^{-2}, 2 \times 10^{-2}, 5 \times 10^{-2}\}$. The number of training epochs is fixed at 20. All experiments are repeated five times, with the mean and standard deviation across the runs reported in Table 5.

Table 5: Comparison of Uni-LoRA and baseline methods on eight computer vision datasets using ViT-Base and ViT-Large backbones. All baseline results, including Full Fine-tuning (FF) and Linear Probing (LP), are taken from the original FourierFT paper [14]. In LP, only the classification head is fine-tuned.

| Model | Method | # Trainable Para. | OxfordPets | StanfordCars | CIFAR10 | DTD | EuroSAT | FGVC | RESISC45 | CIFAR100 | Avg. |
|---|---|---|---|---|---|---|---|---|---|---|---|
| ViT-Base | LP | - | 90.28±0.43 | 25.76±0.28 | 96.41±0.02 | 69.77±0.67 | 88.72±0.13 | 17.44±0.43 | 74.22±0.10 | 84.28±0.11 | 68.36 |
| | FF | 85.8M | 93.14±0.40 | 79.78±1.15 | 98.92±0.05 | 77.68±1.21 | 99.05±0.09 | 54.84±1.23 | 96.13±0.13 | 92.38±0.13 | 86.49 |
| | FourierFT | 72K | 93.21±0.26 | 46.11±0.24 | 98.58±0.07 | 75.09±0.37 | 98.29±0.04 | 27.51±0.64 | 91.97±0.31 | 91.20±0.14 | 77.75 |
| | FourierFT | 239K | 93.05±0.34 | 56.36±0.66 | 98.69±0.06 | 77.30±0.61 | 98.78±0.11 | 32.44±0.99 | 94.26±0.20 | 91.45±0.18 | 80.29 |
| | **Uni-LoRA** | **72K** | **94.00±0.13** | **76.06±0.23** | **98.77±0.03** | **76.99±0.96** | **98.86±0.10** | **50.36±0.63** | 94.08±0.19 | **92.10±0.25** | **85.15** |
| ViT-Large | LP | - | 91.11±0.30 | 37.91±0.27 | 97.78±0.04 | 73.33±0.26 | 92.64±0.08 | 24.62±0.24 | 82.02±0.11 | 84.28±0.11 | 72.96 |
| | FF | 303.3M | 94.43±0.56 | 88.90±0.26 | 99.15±0.04 | 81.79±1.01 | 99.04±0.08 | 68.25±1.63 | 96.43±0.07 | 93.58±0.19 | 90.20 |
| | FourierFT | 144K | 94.46±0.28 | 69.56±0.30 | 99.10±0.04 | 80.83±0.43 | 98.65±0.09 | 39.92±0.68 | 93.86±0.14 | 93.31±0.09 | 83.71 |
| | FourierFT | 480K | **94.84±0.05** | 79.14±0.67 | **99.08±0.05** | **81.88±0.50** | 98.66±0.03 | 51.28±0.66 | 95.20±0.07 | **93.37±0.11** | 86.68 |
| | **Uni-LoRA** | **144K** | 94.65±0.23 | **83.16±0.62** | 98.77±0.03 | 81.35±0.48 | **98.89±0.07** | **58.89±0.62** | **95.24±0.12** | 93.08±0.11 | **88.00** |

Experiments show that Uni-LoRA consistently outperforms FourierFT across computer vision benchmarks. Remarkably, even when FourierFT uses three times as many trainable parameters, Uni-LoRA still achieves better performance. Furthermore, when the number of trainable parameters in Uni-LoRA is less than one-thousandth of that in full fine-tuning, the performance gap between Uni-LoRA and full fine-tuning remains within 2.5%. These results highlight the strong efficiency and generalizability of Uni-LoRA in complex computer vision tasks.

## 4.5 Ablation Study

**Comparison with Fastfood-based Projections.** Fastfood [12] is a classical projection method known for its low computational and memory cost and distance-preserving properties. To evaluate the effectiveness of our projection method, we compare it against Uni-LoRA (Fastfood) on four GLUE tasks in terms of both predictive performance and runtime efficiency. Table 6 shows that Uni-LoRA with the uniform random projection consistently outperforms Uni-LoRA (Fastfood) in terms of predictive performance across four GLUE tasks, while being significantly faster. For example, on MRPC, it achieves a slightly higher score (91.3 vs. 90.7) with a 65% reduction in training time (9 vs.

Table 6: Comparison of Uni-LoRA and Fastfood on four GLUE tasks in predictive performance and training time (mins). We report Matthew's Correlation for COLA and Accuracy for others. Both methods underwent grid search over the same learning rate range, and all other hyperparameters were kept identical across experiments.

| Task | Method | Score (%) | Time (mins) |
|------|--------|-----------|-------------|
| MRPC | Uni-LoRA | 91.3 | 9 |
|      | Fastfood | 90.7 | 26 |
| COLA | Uni-LoRA | 68.5 | 21 |
|      | Fastfood | 65.3 | 60 |
| SST-2 | Uni-LoRA | 96.3 | 80 |
|      | Fastfood | 96.1 | 251 |
| QNLI | Uni-LoRA | 94.6 | 147 |
|      | Fastfood | 94.1 | 358 |

Table 7: Ablation study of Uni-LoRA with local projection and non-uniform projection on four GLUE tasks in predictive performance. For COLA, we report Matthew's correlation and Accuracy for others. All methods were tuned using grid search over the same learning rate range, with all other hyperparameters kept identical across experiments. We report the median performance over 5 runs with different random seeds.

| Task | Uni-LoRA | Local | Non-uniform |
|------|----------|-------|-------------|
| MRPC | $91.3_{\pm0.6}$ | $90.9_{\pm0.8}$ | $90.7_{\pm0.6}$ |
| COLA | $68.5_{\pm1.1}$ | $68.5_{\pm1.3}$ | $67.0_{\pm1.5}$ |
| SST-2 | $96.3_{\pm0.2}$ | $96.2_{\pm0.3}$ | $96.1_{\pm0.4}$ |
| QNLI | $94.6_{\pm0.4}$ | $94.5_{\pm0.1}$ | $94.0_{\pm0.4}$ |

26 mins). This efficiency stems from the linear time complexity $\mathcal{O}(D)$ of our projection, compared to Fastfood's $\mathcal{O}(D \log d)$.

**Comparison with Layer-wise Projections.** To investigate the impact of global vs. layer-wise (local) projections to our method, we construct a controlled variant as follows. On the RoBERTa-large model, we project the parameters of each layer into its own trainable subspace, with each local projection matrix constructed in the same way as that of Uni-LoRA's. To ensure a fair comparison, we set the per-layer subspace dimensionality such that the total number of trainable space dimensionality matches that of Uni-LoRA's. Table 7 shows that local projection performs worse than global projection on MRPC, SST-2, and QNLI, while both methods achieve comparable performance on CoLA.

**Comparison with Non-uniform Projections.** To investigate the impact of uniform vs. non-uniform projections to our method, we construct a controlled variant in which two separate projection matrices are used: one projects all LoRA $A$ matrices into the first two-thirds of the subspace dimensions, and the other projects all $B$ matrices into the remaining one-third. Both projection matrices are constructed using the same method as that of Uni-LoRA's. Table 7 shows that the uniform projection consistently outperforms the non-uniform projection across all the tasks.

## 5  Conclusion

This paper introduces Uni-LoRA, a unified framework of LoRA that reinterprets a broad class of parameter-efficient LoRA variants through the lens of global subspace projection. In light of this unified view, Uni-LoRA further introduces an efficient and theoretically grounded projection matrix, which enjoys all three desired properties of globality, uniformity, and isometry for adaptation performance. Uni-LoRA requires no architectural modifications, no sparsity priors, and achieves a significantly lower time complexity. Our results show that even within the LoRA space, which is already low-rank, there exists an additional low-dimensional parameterization that has several orders of magnitude fewer trainable parameters than LoRA.

**Limitations and broader impacts** As an extreme parameter-efficient fine-tuning method, Uni-LoRA doesn't possess any significant limitations other than it is currently only evaluated on small to medium scale NLP/CV benchmarks due to limited computing resources. As for its broader impacts, Uni-LoRA enables high-quality adaptation of LLMs with minimal computational resources, contributing to the democratization of model customization. We do not foresee any societal risks beyond those generally associated with the use of LLMs.

## Acknowledgments

We would like to thank the anonymous reviewers for their excellent comments and suggestions, which greatly helped improve the quality of this paper. This work was supported in part by the National Science Foundation under Grant Nos. 2343619, 2416872, 2244219, 2315596, and 2146497.

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

# A Appendix

## A.1 Representations of additional LoRA/PEFT methods in the unified framework

**Tied-LoRA/VeRA** As introduced in Section 3.1, Tied-LoRA [3] and VeRA [4] represents the weight increment as $\Delta W = \Lambda_b P_B \Lambda_d P_A$, where $P_B$ and $P_A$ are tied/shared across all the LoRA-adapted modules, and $\Lambda_b$ and $\Lambda_d$ are defined per LoRA module. Specifically, for each LoRA module $\ell = 1, \cdots, L$, we extract the diagonal elements of $\Lambda_b^\ell \in \mathbb{R}^{m \times m}$ and $\Lambda_d^\ell \in \mathbb{R}^{r \times r}$, and concatenate them to construct a trainable parameter vector:

$$\theta_d = \text{Concat}\left(\text{diag}(\Lambda_b^1), \text{diag}(\Lambda_d^1), \cdots, \text{diag}(\Lambda_b^L), \text{diag}(\Lambda_d^L)\right), \tag{5}$$

where $\text{diag}(\cdot)$ denotes the diagonal vector of a matrix, and $d = L(m + r)$ is the number of trainable parameters of Tied-LoRA or VeRA. On the other hand, the full parameter vector of LoRA can be formulated as

$$\theta_D = \text{Concat}\left(\text{vec}_{\text{row}}(\bar{B}^1), \text{vec}_{\text{row}}(\bar{A}^1), \cdots, \text{vec}_{\text{row}}(\bar{B}^L), \text{vec}_{\text{row}}(\bar{A}^L)\right), \tag{6}$$

where $\bar{B}^\ell = \Lambda_b^\ell P_B$ and $\bar{A}^\ell = \Lambda_d^\ell P_A$ correspond to the LoRA parameters $B^\ell$ and $A^\ell$, respectively, and $\text{vec}_{\text{row}}(\cdot)$ denotes the row-wise flatten of a matrix into a column vector.

As illustrated in Figure 2(a), the projection matrix $P$ of Tied-LoRA and VeRA exhibits a structured sparse pattern, composed of block-diagonal components reshaped from $P_B$ and $P_A$. Specifically,

$$P = \text{Diag}\left(\left\{(P_B)_{i,:}^T\right\}_{i=1}^m, \left\{(P_A)_{j,:}^T\right\}_{j=1}^r, \cdots, \left\{(P_B)_{i,:}^T\right\}_{i=1}^m, \left\{(P_A)_{j,:}^T\right\}_{j=1}^r\right), \tag{7}$$

where $(\cdot)_{i,:}^T$ denotes the transpose of the $i$-th row of a matrix, and $\text{Diag}(v_1, v_2, \cdots, v_{L(m+r)})$ constructs a block diagonal matrix from $L(m + r)$ column vectors. Since $P_B$ and $P_A$ are tied/shared cross all the LoRA modules, the block-diagonal components are repeated $L$ times in $P$. Moreover, the main difference between Tied-LoRA and VeRA is the trainability of $P_B$ and $P_A$, i.e., the projection matrix $P$ is trainable in Tied-LoRA and frozen in VeRA. This is indicated by two different colors in the diagram.

**VB-LoRA** decomposes the $B \in \mathbb{R}^{m \times r}$ and $A \in \mathbb{R}^{r \times n}$ matrices of LoRA into fixed-length sub-vectors as follows:

$$\text{vec}_{\text{row}}(B^\ell) = \text{Concat}(u_1^\ell, \cdots, u_{N_B}^\ell), \text{vec}_{\text{row}}(A^\ell) = \text{Concat}(v_1^\ell, \cdots, v_{N_A}^\ell), \forall \ell \in \{1, \cdots, L\}, \tag{8}$$

where $u_i, v_j \in \mathbb{R}^b$ are sub-vectors of length $b$, and $N_B^\ell = \frac{mr}{b}$ and $N_A^\ell = \frac{rn}{b}$ represent the numbers of sub-vectors decomposed from $B^\ell$ and $A^\ell$, respectively.

VB-LoRA learns a shared vector bank $\mathcal{B} = \{\alpha_1, \cdots, \alpha_h\}$, with $\alpha_i \in \mathbb{R}^b$, and the compositional coefficients to generate each sub-vector of $B$s and $A$s. Specifically, each sub-vector is generated from top-$K$ (e.g., $K = 2$) compositional coefficients over $\mathcal{B}$. Since the coefficients are sparse, we can encode them efficiently by $K$ indices over $\mathcal{B}$ and $K$ coefficient values.

As illustrated in Figure 2(b), under our framework, the original LoRA parameter space $\theta_D$ is formed the same as in Eq. 1. The trainable vector $\theta_d$ of VB-LoRA is formed by concatenating all the vectors in $\mathcal{B}$:

$$\theta_d = \text{Concat}(\alpha_1, \alpha_2, \cdots, \alpha_h). \tag{9}$$

The projection matrix $P$ is structured as a block-diagonal matrix, where each diagonal block is of size $b \times b$, matching the dimensionality of vectors in $\mathcal{B}$. Since $K = 2$ in VB-LoRA, two diagonal blocks in a row in $P$ is responsible to reconstruct one sub-vector of $B$s or $A$s, where the location of the diagonal block corresponds to the coefficient index and the diagonal value is the coefficient value. Note that the compositional coefficients are learned in VB-LoRA. Therefore, the locations and values of the diagonal blocks in $P$ are trained along with $\theta_d$, i.e., the vector bank parameters.

**LoRA-XS** represents the weight increment as $\Delta W = P_B \Lambda_R P_A$, where $P_B \in \mathbb{R}^{m \times r}$ and $P_A \in \mathbb{R}^{r \times n}$ are derived from the singular value decomposition (SVD) of the pretrained weight matrix $W$ and are kept frozen during fine-tuning, while $\Lambda_R \in \mathbb{R}^{r \times r}$ is the only trainable component.

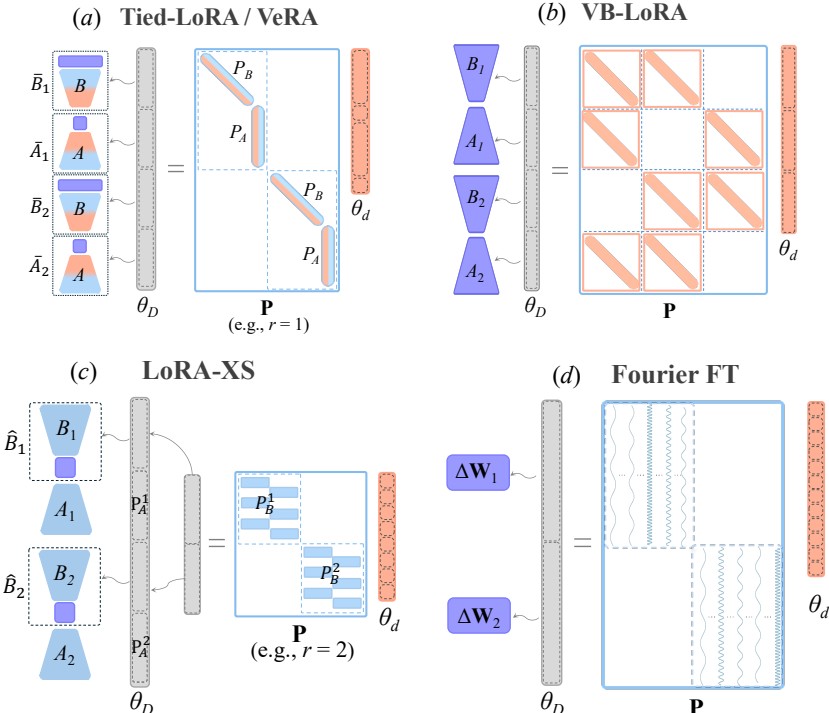

Figure 2: Representations of additional LoRA/PEFT methods in our unified framework. For better visualization, we illustrate the framework with only two LoRA-adapted modules.

Specifically, for each LoRA module $\ell = 1, \cdots, L$, we column-wise flatten each $\Lambda_R^\ell$, and concatenate them to construct a trainable parameter vector:

$$\theta_d = \text{Concat}\left(\text{vec}_{\text{col}}(\Lambda_R^1), \text{vec}_{\text{col}}(\Lambda_R^2), \cdots, \text{vec}_{\text{col}}(\Lambda_R^L)\right), \tag{10}$$

where $\text{vec}_{\text{col}}(\cdot)$ denotes the column-wise flatten of a matrix into a column vector. On the other hand, the full parameter vector of LoRA can be formulated as:

$$\theta_D = \text{Concat}\left(\text{vec}_{\text{row}}(\hat{B}^1), \text{vec}_{\text{row}}(\hat{A}^1), \cdots, \text{vec}_{\text{row}}(\hat{B}^L), \text{vec}_{\text{row}}(\hat{A}^L)\right), \tag{11}$$

where $\hat{B}^\ell = P_B^\ell \Lambda_R^\ell$ and $\hat{A}^\ell = P_A^\ell$ correspond to the $B^\ell$ and $A^\ell$ of LoRA, respectively. Since $\hat{A}^\ell = P_A^\ell$ is initialized, frozen, and independent of $\Lambda_R^\ell$, we only need to map $\Lambda_R^\ell$ to $\hat{B}^\ell$, i.e., projecting the trainable parameter $\theta_d$ to the $\hat{B}$ portion of $\theta_D$. As illustrated in Figure 2(c), the projection matrix $P$ of LoRA-XS also exhibits a structured sparse pattern, composed of the row-vectors extracted from $P_B^\ell$ organized in a stripe pattern.

**FourierFT** isn't a LoRA-based method that performs parameter-efficient fine-tuning in the domain of low-rank adaptation. Instead, FourierFT learns the weight increment $\Delta W$ directly by utilizing the Fast Fourier Transform (FFT) for sparse representation. We show that despite FourierFT isn't a LoRA variant, our Uni-LoRA is a unified framework that can represent FourierFT is the same uniform language and enables a systematic analysis.

FourierFT leverages the Fourier bases to parameterize the weight increments, wherein a small number of trainable Fourier coefficients can be used to synthesize the full weight increments. To represent FourierFT in our unified framework, we row-wise flatten the weight increment $\Delta W^\ell$ for each adapted module $\ell = 1, \cdots, L$, and concatenate them to construct the full PEFT parameter space:

$$\theta_D = \text{Concat}\left(\text{vec}_{\text{row}}(\Delta W^1), \text{vec}_{\text{row}}(\Delta W^2), \ldots, \text{vec}_{\text{row}}(\Delta W^L)\right). \tag{12}$$

On the other hand, the trainable parameter vector $\theta_d$ is formed by concatenating the Fourier coefficients for each adapted module.

As illustrated in Figure 2(d), the projection matrix $P$ also exhibits a block structure since FourierFT adopts a layer-wise projection:

$$P = \text{Diag}\left(\tilde{P}^1, \tilde{P}^2, \cdots, \tilde{P}^L\right),\tag{13}$$

where the block matrix $\tilde{P}^\ell$, denoting the Fourier subspace for the $\ell$-th module, is generated by randomly sampling a subset of Fourier bases.

## A.2 Hyperparameters and Computing Resources

The hyperparameters used for the natural language understanding, mathematical reasoning, and instruction tuning are provided in Tables 8, 9, 10 and 11 . All experiments were conducted on a server equipped with 8 NVIDIA A100 80GB GPUs.

**Computation overhead** The proposed projection in Uni-LoRA is straightforward to implement in modern deep learning frameworks such as PyTorch, enabling full utilization of GPU acceleration. Moreover, since the projection matrix in Uni-LoRA is frozen and does not require training – as opposed to Tied-LoRA and VB-LoRA – it incurs significantly lower computational overhead. As shown in Tables 9 and 10, Uni-LoRA achieves substantially shorter training time compared to VB-LoRA, and is on par with the original LoRA.

**Memory efficiency** Uni-LoRA significantly reduces memory consumption by minimizing the number of trainable parameters. During LoRA fine-tuning, the forward computation is performed as $z = Ax$, $H = Bz$, without the need of explicitly instantiating $\Delta W$. This memory-efficient strategy is seamlessly supported in Uni-LoRA and has been implemented in our codebase. As shown in Tables 9 and 10, Uni-LoRA consistently consumes less memory than VB-LoRA across all model configurations evaluated.

Table 8: Hyperparameters and computing resources used in the natural language understanding experiments on the GLUE benchmark. h: hour, m: minute.

| Model | Hyperparameter | SST-2 | MRPC | CoLA | QNLI | RTE | STS-B |
|---|---|---|---|---|---|---|---|
| | Optimizer | | | AdamW | | | |
| | Warmup Ratio | | | 0.06 | | | |
| | LR Schedule | | | Linear | | | |
| | Init. of $\theta_d$ | | | $\mathcal{U}(-0.02, 0.02)$ | | | |
| BASE | # GPUs | | | 1 | | | |
| | Epochs | 60 | 30 | 80 | 25 | 160 | 80 |
| | Learning Rate (Head) | 1E-4 | 2E-2 | 5E-3 | 2E-4 | 5E-4 | 2E-4 |
| | Learning Rate ($\theta_d$) | | | 5E-3 | | | |
| | $\|\theta_d\|$ | | | 23,040 | | | |
| | Rank | | | 4 | | | |
| | Max Seq. Len. | | | 512 | | | |
| | Batch Size Per GPU | | | 32 | | | |
| | Training Time | 9.2h | 15.5m | 1.8h | 6.2h | 1h | 1.1h |
| | GPU Memory | | | 24,310 MiB | | | |
| LARGE | # GPUs | | | 1 | | | |
| | Epochs | 20 | 40 | 40 | 20 | 40 | 40 |
| | Learning Rate (Head) | 2E-4 | 2E-3 | 2E-2 | 5E-3 | 5E-3 | 1E-4 |
| | Learning Rate ($\theta_d$) | | | 5E-3 | | | |
| | $\|\theta_d\|$ | | | 23,040 | | | |
| | Rank | | | 4 | | | |
| | Max Seq. Len. | | | 128 | | | |
| | Batch Size Per GPU | | | 32 | | | |
| | Training Time | 1.3h | 9m | 21m | 2.5h | 6.5m | 13m |
| | GPU Memory | | | 9,402 MiB | | | |

Table 9: Hyperparameters and computing resources used in the mathematical reasoning experiments. h: hour, m: minute.

| Hyperparameter | Uni-LoRA(Mistral) | Uni-LoRA(Gemma) | VB-LoRA(Mistral) | VB-LoRA(Gemma) |
|---|---|---|---|---|
| # GPUs | | 1 | | |
| Optimizer | | AdamW | | |
| Learning Rate Schedule | | Cosine | | |
| Batch Size | | 1 | | |
| Accumulation Steps | | 64 | | |
| Epochs | | 2 | | |
| Warmup Ratio | | 0.02 | | |
| Rank | | 4 | | |
| Vector bank $b$ | - | - | 256 | 256 |
| Vector bank $h$ | - | - | 2048 | 2048 |
| $|\theta_d|$ | 524,288 | 524,288 | - | - |
| Learning Rate (Vector bank) | - | - | 1E-3 | 1E-3 |
| Learning Rate (Logits) | - | - | 1E-2 | 1E-2 |
| Learning Rate ($\theta_d$) | 2E-3 | 2E-3 | - | - |
| Training Time | 15.5h | 15.1h | 19.3h | 17.5h |
| GPU Memory | 48,984 MiB | 59,488 MiB | 50,426 MiB | 60,166 MiB |

Table 10: Hyperparameters and computing resources used in instruction tuning on the Cleaned Alpaca Dataset. h: hour. 7B: Llama2-7B, 13B: Llama2-13B.

| Hyperparameter | Uni-LoRA-7B | Uni-LoRA-13B | LoRA-7B | LoRA-13B | VB-LoRA-7B | VB-LoRA-13B |
|---|---|---|---|---|---|---|
| # GPUs | | | 1 | | | |
| Optimizer | | | AdamW | | | |
| Warmup Ratio | | | 0.1 | | | |
| Batch Size | | | 4 | | | |
| Accumulation Steps | | | 4 | | | |
| Epochs | | | 1 | | | |
| LR Schedule | | | Linear | | | |
| Rank | 4 | 4 | 64 | 64 | 4 | 4 |
| Vector bank $b$ | - | - | - | - | 256 | 256 |
| Vector bank $h$ | - | - | - | - | 2048 | 4096 |
| $|\theta_d|$ | 524,288 | 1,048,576 | - | - | - | - |
| Learning Rate (Vector bank) | - | - | - | - | 1E-3 | 1E-3 |
| Learning Rate (Logits) | - | - | - | - | 1E-2 | 1E-2 |
| Learning Rate (LoRA) | - | - | 4e-4 | 4e-4 | - | - |
| Learning Rate ($\theta_d$) | 8e-4 | 8e-4 | - | - | - | - |
| Training Time | 3h | 4.5h | 3h | 4.5h | 3.8h | 5.7h |
| GPU Memory | 7,212 MiB | 11,752 MiB | 7,736 MiB | 12,320 MiB | 7,418 MiB | 12,244 MiB |

### A.3 Impact of Hyperparameters on Model Performance

To investigate how the number of trainable parameters $d$ affects model performance, we conducted ablation studies by fine-tuning RoBERTa-Large on the SST-2 task (from GLUE) and Gemma-7B on the mathematical reasoning benchmarks, where $d$ is varied while keeping all other settings at their default values. The results are summarized in Figure 3. Our experiments show that performance improves rapidly with increasing $d$ when $d$ is small, and then plateaus as $d$ grows larger.

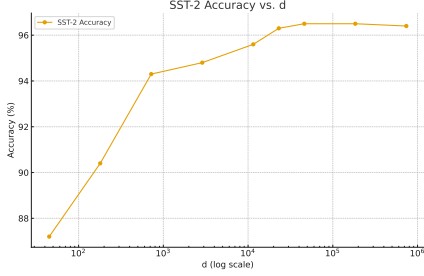

(a) Accuracy on SST-2 as $d$ increases.

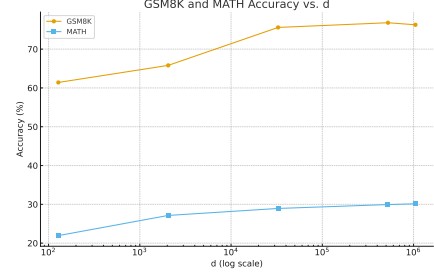

(b) Accuracies on GSM8K and MATH as $d$ increases.

Figure 3: Evolution of Uni-LoRA accuracies on different benchmarks as the number of trainable parameters $d$ increases.

Table 11: Hyperparameters and computing resources used in the vision tasks. h: hour, m: minute.

| Model | Hyperparameter | OxfordPets | StanfordCars | CIFAR10 | DTD | EuroSAT | FGVC | RESISC45 | CIFAR100 |
|---|---|---|---|---|---|---|---|---|---|
| | Optimizer | | | | AdamW | | | | |
| | Weight Decay | | | | 0.01 | | | | |
| | LR Schedule | | | | Linear | | | | |
| | Init. of $\theta_d$ | | | | $\mathcal{U}(-0.02, 0.02)$ | | | | |
| | Epoch | | | | 20 | | | | |
| BASE | # GPUs | | | | 1 | | | | |
| | Learning Rate (Head) | 5E-3 | 5E-2 | 5E-2 | 5E-2 | 5E-2 | 5E-2 | 5E-2 | 1E-2 |
| | Learning Rate ($\theta_d$) | 5E-3 | 1E-2 | 1E-2 | 1E-2 | 1E-2 | 1E-2 | 1E-2 | 1E-2 |
| | $|\theta_d|$ | | | | 72,000 | | | | |
| | Rank | | | | 4 | | | | |
| | Batch Size Per GPU | | | | 128 | | | | |
| | Training Time | 6m | 23m | 49m | 13m | 40m | 53m | 30m | 1.6h |
| | GPU Memory | | | | 12,132 MiB | | | | |
| LARGE | # GPUs | | | | 1 | | | | |
| | Learning Rate (Head) | 2E-2 | 1E-2 | 5E-3 | 1E-2 | 2E-2 | 1E-2 | 1E-2 | 5E-3 |
| | Learning Rate ($\theta_d$) | 1E-2 | 1E-2 | 1E-2 | 1E-2 | 1E-2 | 1E-2 | 1E-2 | 1E-2 |
| | $|\theta_d|$ | | | | 144,000 | | | | |
| | Rank | | | | 4 | | | | |
| | Batch Size Per GPU | | | | 32 | | | | |
| | Training Time | 11m | 33m | 1.6h | 13m | 40m | 53m | 50m | 1.6h |
| | GPU Memory | | | | 30,058 MiB | | | | |

Table 12: All results reported are generated using a consistent version of the GPT-4 API. Score refers to the evaluation on MT-Bench with single-turn dialogues.

| Model | Method | # Parameters | Score | Training Time | GPU memory |
|---|---|---|---|---|---|
| LLAMA2-7B | LoRA (Rank 64) | 159.9M | 5.62 | 3.0 h | 7,736 MiB |
| | LoRA (Rank 4) | 10.0M | 5.34 | 2.9 h | 7,044 MiB |
| | Uni-LoRA (Rank 4) | **0.52M** | 5.58 | 3.0 h | 7,212 MiB |
| LLAMA2-13B | LoRA (Rank 64) | 250.3M | 6.20 | 4.5 h | 12,320 MiB |
| | LoRA (Rank 4) | 15.60M | 6.06 | 4.1 h | 11,634 MiB |
| | Uni-LoRA (Rank 4) | **1.0M** | 6.34 | 4.5 h | 11,752 MiB |

For a fair comparison, we follow VB-LoRA and set the LoRA rank $r = 4$ in the paper. To evaluate the influence of rank $r$, here we conduct additional analysis by varying $r$ for RoBERTa-Large on the SST-2 task (from GLUE) and Gemma-7B on the mathematical reasoning benchmarks. The results reported in Figure 4 show that Uni-LoRA maintains stable performance across a wide range of ranks, with $r = 4$ achieving the best balance between accuracy and efficiency.

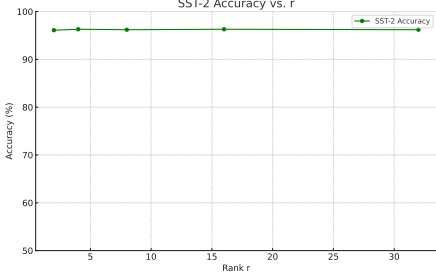

(a) Accuracy on SST-2 with different $r$.

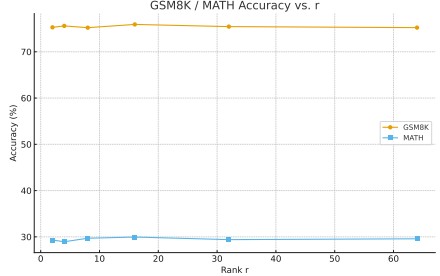

(b) Accuracies on GSM8K and MATH with different $r$.

Figure 4: Performance comparison of Uni-LoRA across different rank $r$.

## A.4 Comparison with LoRA of the Same Rank

In the instruction tuning experiments, we follow the same setup in VeRA [4] and VB-LoRA [6], comparing our method against LoRA with a rank of 64, which is a commonly used setting in prior works. To ensure a fair comparison with our Uni-LoRA, which uses a low rank of 4, we additionally

include a set of instruction tuning experiments where LoRA is also configured with rank 4. These experiments follow the same setup as in Section 4.3, with the only difference being the LoRA rank.

The results are provided in Table 12. It can be observed that reducing the LoRA rank from 64 to 4 leads to a noticeable performance / score drop. Moreover, the performance of the LoRA rank-4 model is consistently worse than that of Uni-LoRA. In contrast, Uni-LoRA employs a fixed, sparse projection matrix, where only the indices and values of the nonzero entries are involved in computation. This leads to an extremely efficient implementation. As a result, Uni-LoRA incurs only marginal increases in training time and memory consumption compared to the rank-4 LoRA baseline, while achieving notable score improvements.

### A.5 Licenses and Asset Usage

We document all external assets used in this work, including models and datasets, along with their licenses and source URLs.

**Natural Language Understanding.** We use RoBERTa-base and RoBERTa-large models developed by Facebook AI, released under the MIT License and available at: `https://huggingface.co/roberta-base`, `https://huggingface.co/roberta-large`. We evaluate on the GLUE benchmark, which is publicly available at `https://gluebenchmark.com/` and composed of multiple sub-datasets under various open licenses, as documented on the GLUE website.

**Mathematical Reasoning.** We fine-tune the Mistral-7B-v0.1 model, released under the Apache 2.0 License and available at `https://huggingface.co/mistralai/Mistral-7B-v0.1`, and the Gemma-7B model, which requires agreement to Google's usage license and is available at `https://huggingface.co/google/gemma-7b`. We use the MetaMathQA dataset, available under the MIT License at `https://huggingface.co/datasets/meta-math/MetaMathQA`, and evaluate on GSM8K and MATH datasets, both under the MIT License and available at: `https://huggingface.co/datasets/openai/gsm8k`, `https://github.com/hendrycks/math`.

**Instruction Tuning.** We use the Cleaned Alpaca dataset, which improves upon the original Alpaca dataset. Both versions are licensed under CC BY-NC 4.0 and available at: `https://huggingface.co/datasets/tatsu-lab/alpaca`, `https://huggingface.co/datasets/yahma/alpaca-cleaned`. We evaluate on MT-Bench, released under CC BY 4.0 and available at `https://huggingface.co/datasets/lmsys/mt_bench_human_judgments`. Fine-tuning is performed on the LLaMA 2 model, licensed under the LLAMA 2 Community License and available at `https://huggingface.co/meta-llama`, using the QLoRA framework, released under the MIT License and available at `https://github.com/artidoro/qlora`.

All assets were used in compliance with their respective licenses. No proprietary or restricted-access data was used in this study.

