# OpenReview forum: "Uni-LoRA: One Vector is All You Need"
_NeurIPS.cc/2025/Conference — NeurIPS 2025 spotlight_

### Official Review · Reviewer_5yKv · 2025-06-10

**Clarity:** 2
**Significance:** 3
**Originality:** 3
**Rating:** 4
**Confidence:** 4

**Summary:**

In this paper, the authors present Uni-LoRA, a unified framework for understanding and improving parameter-efficient fine-tuning (PEFT) methods for large language models (LLMs), particularly those based on Low-Rank Adaptation (LoRA). The key insight is that various LoRA variants, including Tied-LoRA, VeRA, and VB-LoRA, can be reinterpreted through a common lens: all implicitly perform a projection from a low-dimensional subspace \( \mathbb{R}^d \) to the full LoRA parameter space \( \mathbb{R}^D \), where \( d \ll D \). Under this formulation, the primary difference among methods lies in the choice of the projection matrix \( P \in \mathbb{R}^{D \times d} \). The authors further propose an isometric projection matrix that enables global parameter sharing across layers and modules, leading to a more efficient use of trainable parameters. The authors conduct experiments to demonstrate the efficiency of the Uni-LoRA.

**Questions:**

1. How do the authors ensure training stability, particularly given the random initialization of the projection matrix? Does this introduce higher variance or increase the standard error across runs?

2. Can the authors provide results under varying values of r and d to better illustrate the trade-off between parameter efficiency and performance?

3. It would be helpful to include an analysis of GPU memory consumption trends as the model size increases, to understand the scalability of Uni-LoRA in terms of memory footprint.

4. I am curious about the performance of Uni-LoRA on reasoning-focused models. Could the authors provide preliminary results or insights in this area?

**Ethical Concerns:**

["NO or VERY MINOR ethics concerns only"]

**Final Justification:**

My concerns have been addressed by the author during the rebuttal. Therefore, I raised my score.

**Limitations:**

yes

**Quality:**

2

**Strengths And Weaknesses:**

**Strengths**:

1. Presents a novel and unified perspective that subsumes a wide range of LoRA-based extreme PEFT methods under a common framework.

2. Provides both theoretical justification and empirical validation for the proposed projection matrix design.

3. The method is lightweight and easily integrable into existing mainstream fine-tuning frameworks.

**Weaknesses**:

1. The comparisons in Table 3 and Table 4 are limited to only two or three baselines. It would strengthen the evaluation to include methods such as LoRA-XS and FourierFT under comparable trainable parameter budgets.

2. While the proposed projection matrix improves parameter efficiency, it may introduce additional inference latency compared to standard LoRA. An analysis of inference speed would provide a more complete picture of the trade-offs.

3. Although Uni-LoRA significantly reduces the number of trainable parameters, the gains in training time and memory usage are marginal, particularly on larger models (e.g., LLaMA2-7B and LLaMA2-13B, as shown in Table 10). Additional discussion would be beneficial.

---

> ### Author Rebuttal · Authors · 2025-07-30
>
> Dear Reviewer 5yKv,
> ### **Weakness 1 – Limited baselines in Tables 3 and 4**
> Thanks for the suggestion. Tables 3 and 4 initially included only 2–3 baselines because we aimed to include the results reported in their original papers. For Table 3, we have now added two additional baselines: VeRA and FourierFT. When reproducing these baselines, we used their original hyperparameter configurations for fine-tuning.
>
> | Model         | Method     | # Parameters | GSM8K | MATH  |
> |---------------|------------|--------------|-------|-------|
> | MISTRAL-7B    | Full-FT    | 7,242M        | 67.02 | 18.60 |
> |               | LoRA       | 168M         | 67.70 | 19.68 |
> |               | LoRA-XS    | 0.92M        | 68.01 | 17.86 |
> |               | VB-LoRA    | 93M          | 69.22 | 17.90 |
> |               | VeRA       | 1.39M        | 68.69 | 18.81 |
> |               | FourierFT  | 0.67M        | 66.92 | 17.50 |
> |               | **Uni-LoRA**   | **0.52M**        | **68.54** | **18.18** |
> | GEMMA-7B      | Full-FT    | 8,538M        | 71.34 | 22.74 |
> |               | LoRA       | 200M         | 74.90 | 31.28 |
> |               | LoRA-XS    | 0.80M        | 74.22 | 27.62 |
> |               | VB-LoRA    | 113M         | 74.86 | 28.90 |
> |               | VeRA       | 1.90M        | 74.98 | 28.84 |
> |               | FourierFT  | 0.59M        | 72.97 | 25.14 |
> |               | **Uni-LoRA**   | **0.52M**        | **75.59** | **28.94** |
>
> The results demonstrate that Uni-LoRA achieves comparable performance to VeRA on mathematical reasoning tasks, while requiring only **half** as many trainable parameters on Mistral-7B and less than **one-third** on Gemma-7B. As of FourierFT, Uni-LoRA consistently outperforms it across both tasks, while using fewer trainable parameters. Furthermore, Uni-LoRA requires only 15 hours of training time under the same hardware setup, which is significantly faster than FourierFT (73 hours), and comparable to VeRA.
>
> As for Table 4, the experimental setup is based on QLoRA-quantized models. Unfortunately, as stated in their documentation or code comments, the publicly released implementations of LoRA-XS and FourierFT are currently not compatible with quantized base models, so we could not include them as baselines in this setting.
>
> ---
> ### **Weakness 2 – Inference efficiency**
> Thanks for the question. Similar to LoRA, at inference time, Uni-LoRA doesn’t introduce any memory and compute overhead beyond the backbone models since **B** and **A** matrices are merged to the backbone models. The difference between LoRA and Uni-LoRA is that after training, Uni-LoRA regenerates **B** and **A** matrices through $P×θ_d$, and then merges **B** and **A** to the backbone models for inference.
>
> ---
> ### **Weakness 3, Question 3 – GPU memory consumption**
> Thanks for the question. Similar to prior works such as VeRA, VB-LoRA, FourierFT, Tied-LoRA, and LoRA-XS, our focus is on reducing the number of trainable parameters of LoRA. We do not claim reductions in memory or training cost compared to LoRA, and we only claim that Uni-LoRA reduces the storage cost of LoRA parameters significantly. Reducing the number of trainable parameters can substantially decrease both storage and transmission cost for model deployment and is highly valuable in industry. For example, in federated learning (FL), it can significantly lower communication costs while preserving user privacy. We will clarify this further in the manuscript.
>
> ---
> ### **Question 1 – Training stability, particularly given the random initialization**
>
> Thanks for the question. In Table 2 of the main paper, we report the median and standard deviation across five different seeds for each method on six tasks. Below, we present the average of the standard deviations across these six tasks:
>
> | Method     | Average Std (RoBERTa-Base) | Average Std (RoBERTa-Large) |
> |------------|----------------------------|------------------------------|
> | LoRA       | 0.550                      | 0.717                        |
> | VeRA       | 0.417                      | 0.567                        |
> | Tied-LoRA  | 0.850                      | 0.383                        |
> | VB-LoRA    | 0.500                      | 0.717                        |
> | FourierFT  | 0.583                      | 0.677                        |
> | Uni-LoRA   | 0.500                      | 0.677                        |
>
> As can be seen, our method demonstrates comparable stability to other baselines. For RoBERTa-Base, only VeRA has a slightly lower average standard deviation than Uni-LoRA. For RoBERTa-Large, both VeRA and FourierFT have slightly better stability, but Uni-LoRA remains highly competitive in terms of stability. We will clarify this in the main paper.
>
> ---
> ### **Question 2 – Results under varying values of r and d**
> Thanks for the suggestion. We have now added the ablation on r and d. Please refer to our responses to Reviewer FicU, Disadvantage 1, Question 3  and Reviewer 6tip, Weakness 5.
>
> ---
> ### **Question 4 – Uni-LoRA on reasoning-focused models**
> Thanks for the suggestion. We apologize that we couldn’t provide preliminary results on reasoning-focused models given our limited experience on reasoning models and a very tight rebuttal schedule. However, we could provide some insights in this area.
>
> To the best of our knowledge, several recent works have successfully applied LoRA to reasoning-focused models [1-3]. Since Uni-LoRA essentially reconstructs the LoRA parameters from a low parameter space, we believe it can in principle be applied to any models that LoRA performs well (including NLP and CV models, and see our response to Reviewer FicU on Uni-LoRA for CV tasks). It’s interesting to extend Uni-LoRA to the reasoning-focused models, and we plan to explore this direction in the future.
>
> **References:**
>
> [1] Li, Xiang, et al. "MoDE-CoTD: Chain-of-Thought Distillation for Complex Reasoning Tasks with Mixture of Decoupled LoRA-Experts." LREC-COLING  2024.
> [2] Wang, Shangshang, et al. "Tina: Tiny reasoning models via lora." arXiv preprint arXiv:2504.15777 (2025).
> [3] Xue, Yihao, and Baharan Mirzasoleiman. "LoRA is All You Need for Safety Alignment of Reasoning LLMs." arXiv preprint arXiv:2507.17075 (2025).

---

> > ### Comment · Reviewer_5yKv · 2025-08-05
> >
> > Thanks for your response. My concerns have been addressed. Therefore, I'll raise my score accordingly.

---

> > > ### Author Response · Authors · 2025-08-06
> > >
> > > Thank you very much for your thoughtful follow-up and reviewing our responses. We truly appreciate your updated evaluation.

---

### Official Review · Reviewer_k62c · 2025-06-21

**Clarity:** 3
**Significance:** 3
**Originality:** 3
**Rating:** 5
**Confidence:** 4

**Summary:**

The paper presents a general framework that expresses mappings between the trained parameter subspace and the LoRA parameters (usually matrices $A_l$ and $B_l$) of different more efficient LoRA variants (LoRA, Tied-LoRA, VeRA, VB-LoRA, LoRA-XS, FourierFT). The paper expresses the concatenation of all trained and fixed LoRA parameters $\theta_D\in \mathbb R^{D}$ as a product of a projection matrix $P\in\mathbb R^{D\times d}$ (trained or fixed) and a trained lower-dimensional vector $\theta_d\in \mathbb R^d$:
$$
\theta_D = P \theta_d \text.
$$

Based on this expression, where each trained parameter can influence parameters in multiple LoRA modules, the paper proposes Uni-LoRA for greater parameter efficiency. In Uni-LoRA, $P$ is fixed and each of its rows has a single randomly chosen non-zero element. The sparsity enables $O(D)$ time complexity. The value of non-zero elements is set to $P[i, j] = 1/\sqrt{[\![\sum_{j=1}^D P[i, j]\neq 0]\!]}$ to make $P$ isometric (L2 distance preserving).

**Results**

The method is compared with other PEFT methods (LoRA, Tied-LoRA, VeRA, LoRA-XS, and FourierFT) on the GLUE language understanding benchmark (6 tasks) with 2 RoBERTa-base and RoBERTa-large. Uni-LoRA optimizes about 23k (~8% of LoRA) parameters (less than all the other methods) and scores probably among the best 2 PEFT methods.

Uni-LoRA is also very competitive on mathematical reasoning (Mistral-7B and Gemma-7B on GSM8K and MATH) and instruction tuning (LLama2 7B and 13B on MT-Bench) while optimizing the least amount of parameters (0.52M - 1M, 0.26%-0.4% of LoRA).

Ablation experiments on 4 GLUE tasks show that the uniform projection of UniLoRA is more parameter-efficient than non-uniform (twice as much parameters allocated to $A$ matrices than to $B$ matrices) and (maybe) local (per-layer) projections.

**Questions:**

**Questions:**

1. Please provide better support regarding weakness 1.
2. Table 3: Do you have any intuition why LoRA outperforms other methods on MATH, but not GSM8k?

**Ethical Concerns:**

["NO or VERY MINOR ethics concerns only"]

**Final Justification:**

The authors have addressed the most important concerns of all reviewers. The discussion has confirmed my positive score.

**Limitations:**

yes

**Paper Formatting Concerns:**

No concerns.

**Quality:**

4

**Strengths And Weaknesses:**

**Strengths**

1. The presented framework is interesting.
2. The proposed method outperforms or competes very well with other PEFT methods.
3. The paper is very well written.
4. The source code for reproducing the experiments will be published (If I understand correctly.).

**Weaknesses**

1. L202, 233-237: "isometry [...] can substantially enhance adaptation performance". This sounds intuitive, but I don't find it clear how important isometry is. I suggest supporting the claims better.

Minor issues:
- I am unable to find the code for reproducing some experiments (Fastfood).
- LoRA-XS could be described more clearly: SVD is not mentioned.
- L57: Consider clarifying in which way $P$ is theoretically grounded.
- Figure 1: "top-k of learnable logits" is mentioned in the legend, but I don't see it being used.
- L167: I don't find it clear why "Apparently" is used.

---

> ### Author Rebuttal · Authors · 2025-07-30
>
> Dear Reviewer k62c,
> ### **Weakness 1, Question 1 – On the Importance of Isometry**
>
> Thanks for the question. We ensure that the projection matrix **P** in Uni-LoRA is isometric for the following key reasons:
> Yes, standard gradient descent methods are not parameterization-invariant. Poorly scaled parameterizations, which distort the geometric structure of the optimization landscape, can lead to unstable step sizes and slow convergence. Non-isometric transformation can make changes sensitive in one direction and insensitive in another direction. (Please see: J. Nocedal and S. J. Wright, *Numerical Optimization* 2nd ed., Chapters 2.2 and 4.5 for more discussions on the effects of poor scaling.)
>
> Although heuristics such as gradient clipping and gradient-normalized Adam [1] exist as remedies, we prefer to maintain isometry as a desirable property from the outset in reparameterization. This ensures that the conditioning of the optimization problem remains unchanged -- no artificial ill-conditioning is introduced by the reparameterization. We will add this discussion in the updated manuscript.
>
> **References:**
> [1] Yu, Adams Wei, et al. "Block-normalized gradient method: An empirical study for training deep neural network." arXiv preprint arXiv:1707.04822 (2017).
>
> ---
>
> ### Minor Issues
>
> - **Unable to find the code for reproducing some experiments (Fastfood)**
>   For reproducibility, we will open-source all our Uni-LoRA code, including our Fastfood implementation and the Uni-LoRA code for ViT fine-tuning introduced during the rebuttal.
>
> - **LoRA-XS could be described more clearly: SVD is not mentioned.**
>   We apologize for the lack of clarity regarding LoRA-XS. We will clarify that its LoRA-like matrices A and B are derived from the SVD of the pretrained weight matrix **W**.
>
> - **L57: Consider clarifying in which way is theoretically grounded.**
>   Thanks for pointing this out. As we analyzed in Sec. 3.3, **P** possesses three desired properties for adaptation performance, i.e., globality, uniformity/load-balanced, and isometry. We will clarify this at L57 and add a reference to Sec. 3.3.
>
> - **Figure 1: "top-k of learnable logits" is mentioned in the legend, but I don't see it being used.**
>   Thanks for pointing this out. The “top-k of learnable logits” is specific for VB-LoRA, and refers to the orange boxes in the VB-LoRA illustration (Fig. 1d). In this case, *k=2* refers to the two diagonal sub-matrices. We will clarify this in Fig. 1.
>
> - **L167: I don't find it clear why "Apparently" is used.**
>   Thanks for the question. Since the projection matrix **P** in standard LoRA is just an identity matrix, which satisfies *PᵀP = I*, it is inherently isometric.
>
> ---
>
> ### **Question 2 – Any intuition why LoRA outperforms other methods on MATH, but not GSM8k?**
>
> Thanks for pointing this out. This is an interesting observation that we neglected before. We don't have a solid explanation or intuition at this time. We will investigate this in our future work.

---

> ### Comment · Reviewer_k62c · 2025-08-03
>
> Thank you for the clarifications and additional experiments! Having considered all reviews and responses to them, I see no important weakness. Therefore, I will keep my positive assessment of the work.

---

> > ### Author Response · Authors · 2025-08-04
> >
> > Thank you very much for your thoughtful follow-up and for taking the time to carefully read all reviews and our responses. We sincerely appreciate your support to our work.

---

### Official Review · Reviewer_6tip · 2025-06-23

**Clarity:** 1
**Significance:** 2
**Originality:** 2
**Rating:** 4
**Confidence:** 3

**Summary:**

This paper presents Uni-LoRA, a unified framework for Low-Rank Adaptation (LoRA) that generalizes and improves upon existing parameter-efficient fine-tuning (PEFT) methods. The authors claim that their Uni-LoRA achieves state-of-the-art parameter efficiency while
outperforming or matching prior approaches in predictive performance.

**Questions:**

Check the weaknesses part, I'm mainly concerned about 1-5.

**Ethical Concerns:**

["NO or VERY MINOR ethics concerns only"]

**Final Justification:**

While I appreciate the authors' response and have carefully considered their explanations, my initial assessment of the paper's contribution remains unchanged. However, I have adjusted the score upward in recognition of their thorough rebuttal efforts.

**Limitations:**

yes, some limitations are discussed in Conclusion section.

**Paper Formatting Concerns:**

There is no major formatting issues in this paper.

**Quality:**

2

**Strengths And Weaknesses:**

> **Strengths:**
>
> Uni-LoRA advances PEFT by unifying LoRA variants into a single framework, this can help us better understand LoRA-based fine-tuning methods and further reduce the number of parameters required by LoRA.

> **Weaknesses:**
>
> (1) Line 213-214 (Theorem 1) says "P is a projection matrix where each row selects exactly one entry uniformly at random to be nonzero, and sets all other entries to zero". Lacks stability analysis (sampling has randomness).
>
> (2) Theorem 1 is incorrect because it fails to account for cases where all rows of matrix P are sampled identically. For example, when D=3, P = ((1/√3, 1/√3, 1/√3)ᵀ, (0, 0, 0)ᵀ). In this case, clearly PᵀP ≠ I. Additional analysis on the impact of P's rank should be included.
>
> (3) For fine-tuning, LoRA is not shown to reach a comparable performance as full-rank fine-tuning [1]. Further investigation should be conducted on the impact of reducing the number of LoRA parameters.
>
> [1] GaLore: Memory-Efficient LLM Training by Gradient Low-Rank Projection. ICML 2024 Oral.
>
> (4) Lines 329-334: Why does "local projection perform worse than global projection"? Under sufficient training (with other hyperparameters being the same), each local part should learn parameters corresponding to its global counterpart.
>
> (5) Line 304: When comparing LoRA, only the case of rank=4 was evaluated. Different ranks should be compared to better demonstrate the effectiveness of the proposed method.
>
> (6) Lines 319-320: Could this be a formatting issue?

---

> ### Author Rebuttal · Authors · 2025-07-30
>
> Dear Reviewer 6tip,
>
> ### **Weakness 1 – Analysis of Method Stability**
>
> Thanks for the question. In Table 2 of the main paper, we report the median and standard deviation across five different seeds for each method on six tasks. Below, we present the average of the standard deviations across these six tasks:
>
> | Method      | Average Std (RoBERTa-Base) | Average Std (RoBERTa-Large) |
> |-------------|----------------------------|------------------------------|
> | LoRA        | 0.550                      | 0.717                        |
> | VeRA        | 0.417                      | 0.567                        |
> | Tied-LoRA   | 0.850                      | 0.383                        |
> | VB-LoRA     | 0.500                      | 0.717                        |
> | FourierFT   | 0.583                      | 0.677                        |
> | Uni-LoRA    | 0.500                      | 0.677                        |
>
> As can be seen, our method demonstrates comparable stability to other baselines. For RoBERTa-Base, only VeRA has a slightly lower average standard deviation than Uni-LoRA. For RoBERTa-Large, both VeRA and Tied-LoRA have slightly better stability, but Uni-LoRA remains highly competitive in terms of stability. We will clarify this in the main paper.
>
> ---
> ### **Weakness 2  – Clarification on Theorem 1**
>
> We would like to clarify that Theorem 1 assumes $ n_j > 0 $, indicating that each column of the projection matrix **P** must have at least one non-zero entry (see Theorem 1). The counter example provided does not satisfy this assumption.
>
> In practice, the chance to sample a **P** in which one column is entirely zero is extremely low because **P** is a very tall and narrow matrix (where each row has only one non-zero element at a random location). For example, in the LLaMA-2 7B experiments in Sec. 4.3, **P** has a shape of 10M × 0.52M. In this case, the probability that a column is entirely zero is less than 0.23%.
>
> To ensure the theorem’s condition ($n_j > 0$) is always held, we can re-sample **P** if one column happens to be all zeros. We will clarify this further in the main paper.
>
> ---
> ### **Weakness 3  – LoRA vs. FFT and Impact of Reducing the Number of LoRA Parameters**
>
> Numerous studies have shown that LoRA significantly reduces memory and compute cost while achieving comparable or even superior performance to full fine-tuning (FFT) [1-5]. Its efficiency and effectiveness have led to widespread adoption in industry.
>
> Meanwhile, the parameter redundancy of LoRA has been explored in several works, such as VeRA, Tied-LoRA, VB-LoRA, and LoRA-XS. These methods reduce the number of trainable parameters while maintaining similar performance to LoRA, highlighting the parameter redundancy of LoRA. Uni-LoRA builds upon this insight and further improves the parameter efficiency for LoRA.
>
> **References:**
> [1] Hu, Edward J., et al. *"Lora: Low-rank adaptation of large language models."* ICLR 1.2 (2022): 3.
> [2] Dettmers, Tim, et al. *"Qlora: Efficient finetuning of quantized llms."* NeurIPS 36 (2023): 10088-10115.
> [3] Biderman, Dan, et al. *"LoRA Learns Less and Forgets Less."* Transactions on Machine Learning Research.
> [4] Shuttleworth, Reece, et al. *"Lora vs full fine-tuning: An illusion of equivalence."* arXiv preprint arXiv:2410.21228 (2024).
> [5] Lian, Chenyu, et al. *"Less could be better: Parameter-efficient fine-tuning advances medical vision foundation models."* arXiv:2401.12215 (2024).
>
> ---
> ### **Weakness 4  – Why Does Local Projection Perform Worse Than Global Projection?**
> Thanks for the question. As highlighted by AdaLoRA [1], when fine-tuning with LoRA, not all layers contribute equally, i.e., the importance and information content of each LoRA module varies. Prior methods such as VeRA, Tied-LoRA, and LoRA-XS use local projections with the same projection dimension for each layer, which means layers with different information content are projected into different subspaces of the same size. This same-sized treatment for each layer is suboptimal. In contrast, global projection projects all LoRA parameters into a shared subspace to mitigate the aforementioned issue.
>
> Local projection is analogous to pouring five unequal volumes of water into five identical cups, which may lead to overflow in some cups and under-utilize others. In contrast, global projection is analogous to pouring all water into a single container, which allows more efficient utilization of capacity. The ablation studies in Table 6 support this claim. We will clarify this further in the updated manuscript.
>
> **References:**
> [1] Zhang, Qingru, et al. *"Adaptive Budget Allocation for Parameter-Efficient Fine-Tuning."* ICLR 2023.
>
> ---
> ### **Weakness 5  – Additional Analysis on the Influence of Rank *r***
>
> Thanks for the suggestion. For a fair comparison, in the paper we followed VB-LoRA and set the LoRA rank *r* = 4. To evaluate the influence of rank *r*,  here we conducted additional analysis by varying *r* for RoBERTa-Large on the SST-2 task (from GLUE) and Gemma-7B on the mathematical reasoning benchmarks. The results reported in the table below show that Uni-LoRA maintains stable performance across a wide range of ranks, with *r* = 4 achieving the best balance between accuracy and efficiency. We will include this ablation study in the updated manuscript.
>
> SST-2 Accuracy vs. *r*
>
> | Rank *r*         | 1        | 2        | 4        | 8        | 16       | 32       |
> |------------------|----------|----------|----------|----------|----------|----------|
> | SST-2 ACC        | 89.2±1.4 | 96.1±1.4 | 96.3±0.2 | 96.2±0.1 | 96.3±0.2 | 96.2±0.1 |
>
>
>
>  GSM8K / MATH Accuracy vs. *r*
>
> | Rank *r*         | 2       | 4       | 8       | 16      | 32      | 64      |
> |------------------|---------|---------|---------|---------|---------|---------|
> | GSM8K ACC        | 0.7528  | 0.7559  | 0.7521  | 0.7591  | 0.7544  | 0.7522  |
> | MATH ACC         | 0.2932  | 0.2894  | 0.2970  | 0.2996  | 0.2940  | 0.2958  |
>
> ---
> ### **Weakness 6  – formatting issue**
> Thanks for pointing this out. We will fix it in the updated manuscript.

---

> > ### Comment · Reviewer_6tip · 2025-08-05
> >
> > Thanks the authors for their response.
> >
> > Several issues remain to be considered by the authors. For example,
> >
> > in your "Weakness 2 – Clarification on Theorem 1".  I see "each column of the projection matrix P must have at least one non-zero entry", but I mean that —— **if P is not full-rank**, taking P = ((1/√3, 1/√3, 1/√3)ᵀ, (1/√3, 1/√3, 1/√3)ᵀ) as an instance, PᵀP ≠ I holds, which implies that Theorem 1 lacks rigor.
> >
> > in your "Weakness 5 – Additional Analysis on the Influence of Rank r". Different ranks should be compared to better demonstrate the effectiveness of the proposed method——the authors should conduct a comparative analysis of different fine-tuning methods across varying values of r.

---

> > > ### Author Response · Authors · 2025-08-05
> > >
> > > ### **Question on P is not full rank**
> > >
> > > Thanks for your follow-up question. First of all, we want to emphasize that Theorem 1 indicates that **P is a full-rank matrix**. This is because each **row** of P has exactly one non-zero entry (chosen uniformly at random) and the assumption of $n_j>0$ make sures that each **column** of P must have at least one non-zero entry. Under the conditions of Theorem 1, P is a full-rank matrix, which is proved below.
> > >
> > > **Proof**: Given any column $i$ of P, denote its non-zero entries as being located at rows $[i_1, i_2, \dots]$. Due to the construction of P that each row contains exactly one non-zero entry, all other columns (all j != i) must have zeros at their rows $[i_1, i_2, \dots]$. Therefore, any column of P cannot be represented as a linear combination of the remaining columns, implying that all columns are linearly independent. Hence, P has a full-column rank $d$. Since the number of rows exceeds the number of columns ($D > d$), $P$ is full rank.
> > >
> > > We will clarify this further in the manuscript.
> > >
> > > ---
> > >
> > > ### **Comparative analysis of different fine-tuning methods across varying values of $r$**
> > >
> > > Thanks for clarifying this question. We are doing additional comparative analysis of different fine-tuning methods across varying $r$. However, we’d like to emphasize that for LoRA, VeRA, Tied-LoRA, and VB-LoRA, the number of trainable parameters linearly increases with rank r, while for Uni-LoRA, the number of trainable parameters $d$ and rank $r$ are decoupled and are two separate hyperparameters. We believe decoupling the trainable parameter count from rank $r$ gives Uni-LoRA additional flexibility.
> > >
> > > It takes some time to conduct these experiments. We will try our best and share the results before the discussion deadline.

---

> > > > ### Comment · Reviewer_6tip · 2025-08-07
> > > >
> > > > Thanks the authors for their response. I have adjusted the score upward in recognition of the thorough rebuttal efforts.

---

> > > > > ### Author Response · Authors · 2025-08-09
> > > > >
> > > > > We sincerely appreciate your thoughtful questions, and we are grateful for your recognition of our rebuttal efforts and the upward score adjustment.

---

> > > ### Author Response · Authors · 2025-08-08
> > >
> > > Thanks for your patience. Here are the results of different fine-tuning methods (**VeRA**, **VB-LoRA**, and **Uni-LoRA**) on the SST-2 task (from GLUE) by varying `r` for RoBERTa-Large. Since the number of trainable parameters (`#trainable paras`) in **VeRA** and **VB-LoRA** increases linearly with the rank `r`, while Uni-LoRA's `#trainable params d` is a hyperparameter, for a fair comparison we additional introduce **Uni-LoRA (match)**, in which both `r` and `#trainable paras` are matched to those of **VeRA**.
> > >
> > > For each method, we report **ACC / #trainable params**. Additionally, for **VB-LoRA**, we distinguish between the number of trainable parameters and the number of saved parameters, and report **ACC / #trainable params / #saved params**.
> > >
> > > All reported **ACC** values represent the **median and standard deviation** across **five runs** with different random seeds.
> > >
> > > | **r**                  | **2**                              | **4**                              | **16**                              | **32**                                | **64**                                | **256**                                 | **512**                                 |
> > > |------------------------|------------------------------------|------------------------------------|-------------------------------------|----------------------------------------|----------------------------------------|------------------------------------------|------------------------------------------|
> > > | **VeRA**               | 96.0±0.13 / 49.2k                 | 95.7±0.36 / 49.3k                 | 95.6±0.39 / 49.9k                  | 95.8±0.32 / 50.7k                     | 95.4±0.56 / 52.2k                     | 95.9±0.16 / 61.4k                       | 95.8±0.13 / 73.7k                       |
> > > | **VB-LoRA**           | 96.0±0.3 / 92.20k / 23.8k         | 96.2±0.2 / 161.3k / 24.6k        | 96.2±0.2 / 576.0k / 29.2k         | 96.0±0.4 / 1,128.9k / 35.3k         | 96.0±0.3 / 2,234.9k / 47.6k         | 95.7±0.2 / 8,870.4k / 121.3k           | 95.4±0.3 / 17,717.8k / 219.6k          |
> > > | **Uni-LoRA (match)**   | 95.8±0.2 / 49.2k                  | 95.8±0.3 / 49.3k                  | 96.3±0.4 / 49.9k                   | 96.0±0.4 / 50.7k                      | 96.1±0.2 / 52.2k                      | 96.1±0.3 / 61.4k                        | 96.3±0.3 / 73.7k                        |
> > > | **Uni-LoRA**           | 96.1±0.1 / 23.0k                  | 96.3±0.2 / 23.0k                  | 96.3±0.2 / 23.0k                   | 96.2±0.1 / 23.0k                      | 96.3±0.2 / 23.0k                      | 96.2±0.1 / 23.0k                        | 96.0±0.1 / 23.0k                        |
> > >
> > >
> > >
> > > ### Key Observations
> > >
> > > 1. **VeRA**, **VB-LoRA**, and **Uni-LoRA** are all very stable with varying `r`, and typically reach a good ACC when `r=4`, which leads to a small `#trainable params` or parameter efficiency. Among them, **Uni-LoRA** is the most parameter efficient one (23k).
> > >
> > > 2. **Uni-LoRA** and **Uni-LoRA (match)** are very stable with varying `r` when `#trainable params d` is fixed to 23k or when `#trainable params d` matches that of **VeRA**. This result shows that increasing `r` doesn't mean we need to increase `#trainable params`. We believe decoupling `#trainable params d` from rank `r` gives Uni-LoRA additional flexibility.
> > >
> > > We will include this result and discussion in the manuscript.

---

### Official Review · Reviewer_FicU · 2025-07-05

**Clarity:** 3
**Significance:** 3
**Originality:** 3
**Rating:** 4
**Confidence:** 4

**Summary:**

Uni-LoRA proposes a unified framework that maps various LoRA variants into a shared low-dimensional subspace via a global projection matrix, enabling parameter sharing and compression. Its core innovation lies in leveraging a single trainable vector to achieve parameter sharing while preserving the isometric structure of the parameter space. This approach significantly reduces memory and computational overhead, while achieving performance comparable to traditional fine-tuning methods on tasks such as GLUE, GSM8K, and MATH.

**Questions:**

1. The distance maintained by the projection matrix P is Euclidean, while most data typically resides in non-Euclidean spaces. Could the adoption of geodesic distance on manifolds be considered to optimize the projection matrix P?
2. The projection matrix P is relatively simple, and it is unclear whether it can maintain good performance on more complex data. Could the use of multiple projection matrices be considered to achieve better performance?
3. Could a discussion on the dimension d of the low-dimensional subspace be included in the parameter experiments to explore the impact of different dimension choices on performance?

**Ethical Concerns:**

["NO or VERY MINOR ethics concerns only"]

**Final Justification:**

Thank you for the clarifications and additional experiments. After carefully considering all reviews and the authors' responses, I find no major weaknesses remaining. I therefore keep my positive assessment of this work.

**Limitations:**

1. Despite the outstanding performance of Uni-LoRA on NLP tasks, its effectiveness in cross-domain applications or complex tasks (such as image processing and cross-lingual transfer) has yet to be fully validated. The authors could further discuss the method's adaptability in other domains, such as computer vision or reinforcement learning, and highlight potential challenges related to its adaptability.
2. The experimental validation in the paper primarily focuses on medium- to large-scale models (e.g., Gemma-7B, Mistral-7B), Uni-LoRA: One Vector is All You Need with limited testing on extremely large models (such as GPT-3). There is insufficient discussion on whether the method can effectively operate on ultra-large models, particularly in distributed training environments.

**Quality:**

3

**Strengths And Weaknesses:**

Advantages:
1. Uni-LoRA demonstrates high computational and memory efficiency in the application of large-scale pre-trained models. The experiments are sufficiently conducted and are backed by a solid theoretical foundation.
2. The structure of the paper is clear, effectively conveying the core ideas of the research, with a well-defined experimental design. The paper provides reproducible experimental procedures and code support.
3. Uni-LoRA reduces the number of parameters while maintaining good performance, making it highly significant for practical applications, particularly in resource-constrained environments.
4. The paper introduces a unified framework that integrates various LoRA variants through a global projection matrix, allowing all variants to be optimized via a shared low-dimensional subspace, thus reducing computational complexity. Additionally, the proposed projection matrix P ensures that the geometric structure of the optimization space remains preserved.

Disadvantages:
1. The details of the projection matrix selection are not fully addressed. While the projection matrix P is described as globally shared and optimized via backpropagation, the paper lacks sufficient discussion on how to dynamically adjust and select the optimal dimension d in practical applications.
2. While the unified framework proposed in the paper improves both parameter and computational efficiency, the simplification of the method may limit its adaptability in certain tasks, especially in more complex scenarios. The effectiveness of a single projection matrix in maintaining good performance for complex tasks warrants further investigation.

---

> ### Author Rebuttal · Authors · 2025-07-30
>
> Dear Reviewer FicU，
> ### **Disadvantage 1, Question 3 – Impact of Projection Dimension *d* on Model Performance**
>
> Thanks for the suggestion. First, we would like to clarify that our projection matrix **P** is not optimized via backpropagation. Instead, **P** is randomly sampled and fixed (see Theorem 1). Since **P** is not learned, our method incurs **lower memory and computational overhead** compared to the baseline methods, such as VBLoRA and Tied-LoRA, which require optimizing projection parameters.
>
> To investigate the impact of *d* on model performance, we conducted ablation studies by fine-tuning RoBERTa-Large on the SST-2 task (from GLUE) and Gemma-7B on the mathematical reasoning benchmarks, where *d* is varied while keeping all other settings at their default values. The results are summarized in the tables below:
>
> #### SST-2 Accuracy vs. *d*
>
> | Dim *d*   | 45   | 180  | 720  | 2,880 | 11,520 | 23,040 | 46,080 | 184,320 | 737,280 |
> |-----------|------|------|------|------|--------|--------|--------|--------|--------|
> | Accuracy  | 87.2±0.2 | 90.4±0.4 | 94.3±0.4 | 94.8±0.2 | 95.6±0.2 | 96.3±0.2 | 96.5±0.3 | 96.5±0.2 | 96.4±0.3 |
>
> #### GSM8K / MATH Accuracy vs. *d*
>
> | Dim *d*   | 128  | 2,048 | 32,768 | 524,288 | 1,048,576 | 2,097,152 |
> |-----------|------|------|--------|--------|----------|-----------|
> | GSM8K ACC | 0    | 0.6141 | 0.6581 | 0.7559 | 0.7680   | 0.7627    |
> | MATH ACC  | 0.0002 | 0.2194 | 0.2714 | 0.2894 | 0.2994   | 0.3014    |
>
> Our experiments show that performance improves rapidly with increasing *d* when *d* is small, and then plateaus as *d* grows larger. We will include these results in the updated manuscript.
>
> ---
>
> ### **Disadvantage 2, Question 2, Limitation 1 – Performance on More Complex Tasks**
>
> To evaluate Uni-LoRA in more complex tasks, we followed the FourierFT setup for computer vision and conducted additional experiments with the results reported in the table below.
>
> Specifically, we applied Uni-LoRA with *d* = 74,000 on ViT-Base and *d* = 144,000 on ViT-Large, for rank *r* = 4. A  grid search was conducted over:
>
> - head learning rate: {1e-3, 2e-3, 5e-3, 1e-2}
> - trainable vector $\theta_d$ learning rate: {2e-3, 5e-3, 1e-2, 2e-2, 5e-2}
>
> while fixing the number of training epochs to 10.
>
> Each experiment was repeated five times, and we reported the mean and standard deviation across the five runs.
>
> >All  results, except those for Uni-LoRA, are taken from the original FourierFT paper, including Full Fine-tuning (FF) and Linear Probing (LP), in which only the classification head is fine-tuned.
>
> | Model     | Method           | # Trainable Parameters | OxfordPets       | StanfordCars     | CIFAR10          | DTD              | EuroSAT          | FGVC             | RESISC45         | CIFAR100         | Avg.   |
> |-----------|------------------|------------------------|------------------|------------------|------------------|------------------|------------------|------------------|------------------|------------------|--------|
> | ViT-Base  | LP               | -                      | 90.28±0.43       | 25.76±0.28       | 96.41±0.02       | 69.77±0.67       | 88.72±0.13       | 17.44±0.43       | 74.22±0.10       | 84.28±0.11       | 68.36  |
> |           | FF               | 85.8M                  | 93.14±0.40       | 79.78±1.15       | 98.92±0.05       | 77.68±1.21       | 99.05±0.09       | 54.84±1.23       | 96.13±0.13       | 92.38±0.13       | 86.49  |
> |           | LoRA             | 581K                   | 93.19±0.36       | 45.38±0.41       | 98.78±0.05       | 74.95±0.40       | 98.44±0.15       | 25.16±0.16       | 92.70±0.18       | 92.02±0.12       | 77.58  |
> |           | FourierFT   | 72K                    | 93.21±0.26       | 46.11±0.24       | 98.58±0.07       | 75.09±0.37       | 98.29±0.04       | 27.51±0.64       | 91.97±0.31       | 91.20±0.14       | 77.75  |
> |           | FourierFT  | 239K                   | 93.05±0.34       | 56.36±0.66       | 98.69±0.06       | 77.30±0.61       | 98.78±0.11       | 32.44±0.99       | 94.26±0.20       | 91.45±0.18       | 80.29  |
> |           | **Uni-LoRA**         | **72K**                    | **92.04±0.14**       | **63.26±0.45**       | **98.80±0.06**       | **75.76±0.48**       | **98.52±0.10**       | **30.08±0.51**       | **93.27±0.22**       | **92.32±0.07**       | **80.50**  |
> | ViT-Large | LP               | -                      | 91.11±0.30       | 37.91±0.27       | 97.78±0.04       | 73.33±0.26       | 92.64±0.08       | 24.62±0.24       | 82.02±0.11       | 84.28±0.11       | 72.96  |
> |           | FF               | 303.3M                 | 94.43±0.56       | 88.90±0.26       | 99.15±0.04       | 81.79±1.01       | 99.04±0.08       | 68.25±1.63       | 96.43±0.07       | 93.58±0.19       | 90.20  |
> |           | LoRA             | 1.57M                  | 94.82±0.09       | 73.25±0.36       | 99.13±0.05       | 81.79±0.45       | 98.63±0.07       | 42.32±0.98       | 94.71±0.25       | 94.87±0.10       | 84.94  |
> |           | FourierFT  | 144K                   | 94.46±0.28       | 69.56±0.30       | 99.10±0.04       | 80.83±0.43       | 98.65±0.09       | 39.92±0.68       | 93.86±0.14       | 93.31±0.09       | 83.71  |
> |           | FourierFT  | 480K                   | 94.84±0.05       | 79.14±0.67       | 99.08±0.05       | 81.88±0.50       | 98.66±0.03       | 51.28±0.66       | 95.20±0.07       | 93.37±0.11       | 86.68  |
> |           | **Uni-LoRA**         | **144K**                  | **94.61±0.09**       | **77.53±0.08**       | **99.10±0.03**      | **79.33±0.52**       | **98.63±0.09**       | **46.51±0.49**       | **93.54±0.13**       | **93.72±0.13**       | **85.37**  |
>
>
> Experiments show that given the same number of trainable parameters Uni-LoRA significantly outperforms FourierFT across computer vision benchmarks. Moreover, when the number of parameters in FourierFT is three times that of Uni-LoRA, their performances on computer vision benchmarks are comparable. These results demonstrate the strong generalizability of Uni-LoRA to complex tasks.
>
> Our core observation is that tasks (NLP or CV) that benefit from LoRA also tend to perform well with Uni-LoRA. This is because Uni-LoRA essentially reconstructs the LoRA parameters from a low parameter space via $P\times\theta_d$. We will incorporate these CV results into the main paper and release all the code for reproducibility.
>
> Regarding multiple projection matrices, this is a very interesting idea, and we plan to explore it in our future work.
>
>
> ---
> ### **Question1 – Consider geodesic distance on manifolds to optimize P?**
>
> Thank you for the suggestion. First of all, as we discussed above, the projection matrix **P** is not optimized. Instead, **P** is randomly sampled and fixed (see Theorem 1). Since **P** is not learned, the number of trainable and stored parameters of Uni-LoRA is extremely small (i.e., $|\theta_d|$ + 1 random seed). Optimizing **P** would increase the number of trainable and stored parameters dramatically and incur large memory and computational overhead.
>
> If we ignore the memory and computational overhead and optimize **P** in the non-Euclidean space, it might be possible to replace Adam with its Riemannian adaptive counterparts [1, 2]. Adapting to the manifold setting, however, requires computing the function’s gradient and, in some cases, the Hessian matrix -- which is computationally prohibitive. The practicality of methods and libraries such as Geoopt [2] and Pymanopt [3] for large-scale deep learning optimization remains uncertain.
>
> **References:**
> [1] Bonnabel, Silvere. *"Stochastic gradient descent on Riemannian manifolds."* IEEE Transactions on Automatic Control 58.9 (2013): 2217-2229.
> [2] Bécigneul, Gary, and Octavian-Eugen Ganea. *"Riemannian Adaptive Optimization Methods."* International Conference on Learning Representations (ICLR 2019). Vol. 9. Curran, 2023.
> [3] Townsend, James, Niklas Koep, and Sebastian Weichwald. *"Pymanopt: A python toolbox for optimization on manifolds using automatic differentiation."* Journal of Machine Learning Research 17.137 (2016): 1-5.
>
> ---
> ### **Limitation 2 – Scalability in Distributed Systems**
>
> Thank you for the suggestion. We apologize that we were unable to fine-tune on GPT-3 (175B) due to resource constraints. However, Uni-LoRA naturally supports data parallelism, and indeed, the results reported in Table 3 were obtained using multi-GPU data-parallel training.
>
> To enable model parallel training of Uni-LoRA across multiple GPUs, we can adopt a strategy where each GPU maintains a local copy of the trainable vector $ \theta_d $, and synchronization is achieved via an all-reduce operation on the gradients during backpropagation. This approach is highly efficient for Uni-LoRA, as the number of trainable parameters is extremely small, resulting in negligible communication overhead across devices.
>
> We would like to emphasize that the extremely small number of trainable parameters constitutes a key advantage of Uni-LoRA over other PEFT methods. For example, when fine-tuning LLaMA2-13B, Uni-LoRA requires synchronizing only 1M parameters, whereas VB-LoRA may require synchronization of up to 256M parameters in the worst case. We will include this discussion in the updated manuscript.

---

> > ### Comment · Reviewer_FicU · 2025-08-07
> >
> > Thank you for the clarifications and additional experiments. After carefully considering all reviews and the authors' responses, I find no major weaknesses remaining. I therefore keep my positive assessment of this work.

---

> > > ### Author Response · Authors · 2025-08-09
> > >
> > > We sincerely thank you for your positive evaluation and your continued support of our work.

---

### Decision · Program_Chairs · 2025-09-17

**Decision:**

Accept (spotlight)

**Comment:**

This paper presents Uni-LoRA, a unified framework for Low-Rank Adaptation (LoRA) that generalizes and improves upon existing parameter-efficient fine-tuning (PEFT) methods. Uni-LoRA demonstrates high computational and memory efficiency in the application of large-scale pre-trained models. The experiments are sufficiently conducted and are backed by a solid theoretical foundation. The reviews are consistently positive. The topic is interesting.